# Crystal structure of an RNA-cleaving DNAzyme

Hehua Liu[1,2], Xiang Yu[1,2], Yiqing Chen[1], Jing Zhang[1], Baixing Wu [2], Lina Zheng[2], Phensinee Haruehanroengra[3], Rui Wang[3], Suhua Li[1], Jinzhong Lin[2], Jixi Li[1], Jia Sheng[3], Zhen Huang[4,5], Jinbiao Ma [2] & Jianhua Gan[1]

In addition to storage of genetic information, DNA can also catalyze various reactions. RNA-cleaving DNAzymes are the catalytic DNAs discovered the earliest, and they can cleave RNAs in a sequence-specific manner. Owing to their great potential in medical therapeutics, virus control, and gene silencing for disease treatments, RNA-cleaving DNAzymes have been extensively studied; however, the mechanistic understandings of their substrate recognition and catalysis remain elusive. Here, we report three catalytic form 8–17 DNAzyme crystal structures. 8–17 DNAzyme adopts a V-shape fold, and the $Pb^{2+}$ cofactor is bound at the pre-organized pocket. The structures with $Pb^{2+}$ and the modification at the cleavage site captured the pre-catalytic state of the RNA cleavage reaction, illustrating the unexpected $Pb^{2+}$-accelerated catalysis, intrinsic tertiary interactions, and molecular kink at the active site. Our studies reveal that DNA is capable of forming a compacted structure and that the functionality-limited bio-polymer can have a novel solution for a functional need in catalysis.

[1] State Key Laboratory of Genetic Engineering, Collaborative Innovation Center of Genetics and Development, Department of Physiology and Biophysics, School of Life Sciences, Fudan University, Shanghai 200438, China. [2] State Key Laboratory of Genetic Engineering, Collaborative Innovation Center of Genetics and Development, Department of Biochemistry, Institute of Plant Biology, School of Life Sciences, Fudan University, Shanghai 200438, China. [3] Department of Chemistry and The RNA Institute, University at Albany, State University of New York, Albany, NY 12222, USA. [4] Department of Chemistry, Georgia State University, Atlanta, GA 30303, USA. [5] College of Life Sciences, Sichuan University, Chengdu, 610041, China. Hehua Liu and Xiang Yu contributed equally to this work. Correspondence and requests for materials should be addressed to Z.H. (email: huang@gsu.edu) or to J.M. (email: majb@fudan.edu.cn) or to J.G. (email: ganjhh@fudan.edu.cn)

In addition to storage of genetic information, DNA can also catalyze various reactions, such as cleavage of RNA[1–3] and DNA[4], ligation of DNA[5] and RNA[6,7], formation of RNA branch[8] and RNA lariat[9], and DNA phosphorylation[10]. Because of their uniqueness in the catalysis and sequence space, catalytic DNAs (DNA catalyst, DNA enzyme, or DNAzyme) have attracted tremendous attention from many research fields. However, only very limited structural information is available. RNA-cleaving DNAzymes (the first group of catalytic DNAs) were discovered more than two decades ago. Besides their great potentials in controlling viruses and silencing genes associated with human diseases[11–15], RNA-cleaving DNAzymes can also serve as useful tool in many research fields, such as fragmentation of natural mRNAs for subsequent analysis[16]; therefore, the RNA-cleaving DNAzymes have attracted much more attention than all other DNAzymes. However, the early attempts to determine structures of the RNA-cleaving 10–23 DNAzyme ended up with the catalytically irrelevant structure at 3.0 Å resolution[17]; the mechanistic understandings of their substrate recognition and catalysis at atomic level remain elusive.

Besides 10–23 DNAzyme, 8–17 DNAzyme can also efficiently catalyze the cleavage reaction of RNA substrate[2,3,18], forming the 5′-product with a 2′,3′-cyclic phosphate at the 3′-end and the 3′-product with a hydroxyl group at the 5′-end; similar structural features are also observed on the cleavage products of the naturally occurred hammerhead, hairpin, and hepatitis delta virus (HDV) ribozymes[19–22]. 8–17 DNAzyme contains a small

catalytic core, which is only 15 nucleotides (nt) in length. Interestingly, the same catalytic motif of 8–17 DNAzyme has been independently isolated under different selection conditions and using different cation cofactors[2,3,23], indicating that this small DNAzyme is efficient and its catalytic core motif is ubiquitous in the sequence space. The central catalytic core of 8–17 DNAzyme is flanked by two substrate-recognizing arms, which bind to the RNA substrate via the canonical Watson–Crick base pairing (Fig. 1a).

Although it was initially isolated by in vitro selection strategy in the presence of $Mg^{2+}$[3], later studies revealed that 8–17 DNAzyme is more active in the presence of some transition metal cations, such as $Zn^{2+}$[2]. Among all the tested cations, it was found that $Pb^{2+}$ works the best with this DNA catalyst, catalyzing the RNA cleavage reaction approximately 200-fold faster than $Mg^{2+}$[24,25]. The estimated $Pb^{2+}$-binding, $Zn^{2+}$-binding, and $Mg^{2+}$-binding affinities ($K_d$) of 8–17 DNAzyme are 0.71, 52.6, and 1360 µM, respectively[18,24,26]; because of its high specificity for $Pb^{2+}$, the 8–17 DNAzyme is also used as a lead sensor[27]. However, it is puzzling how this short DNA sequence folds to generate a catalyst that efficiently cleaves RNA. Moreover, it is intriguing what the catalytic roles of the cation cofactors are and how $Pb^{2+}$ is bound at the active site and catalyzes the cleavage reaction. Here, we report three catalytic form 8–17 DNAzyme crystal structures; the structures with $Pb^{2+}$ and the modification at the cleavage site captured the RNA cleavage reaction at pre-catalytic state. Besides the overall folding and intrinsic tertiary interactions, these structures also

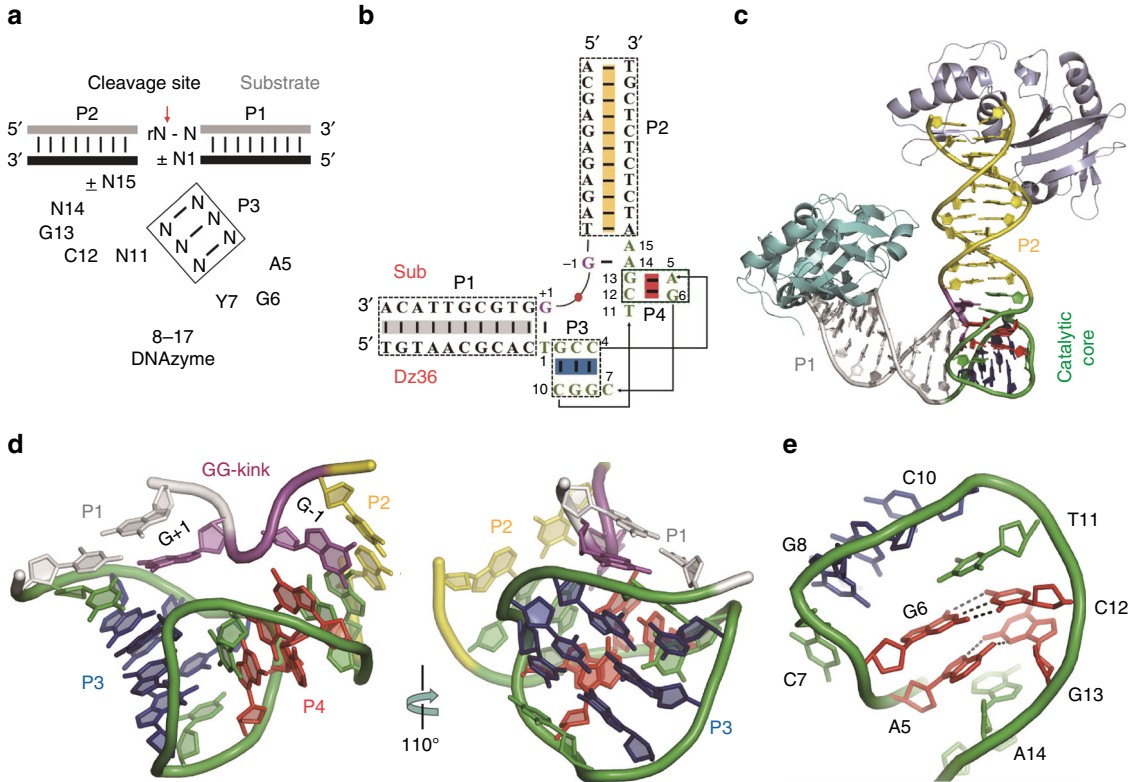

**Fig. 1** Global architecture of the 8–17 DNAzyme. **a** 8–17 consensus sequence and the predicted secondary structure. The sequence was adapted from the commonly observed sequence variations and mutagenesis experiments. **b** Sequence and the observed secondary structure of DNAzyme (Dz36) and the substrate (or analog) utilized in the structural and the catalytic cleavage studies. **c** Cartoon representation showing the overall fold of the DNAzyme/substrate analog complex, based on the high-resolution DNAzyme-$Pb^{2+}$ structure. *Asfv*PolX proteins, which were utilized to facilitate the crystallization, are colored in light blue or cyan. **d** The relative arrangement of P1, P2, GG-kink, and the catalytic core. **e** Close-up view showing the detailed base pairing of the P4 duplex. In **c**–**e**, P1, P2, and GG-kink are colored in white, yellow, and magenta, respectively. The sugar puckers and nucleobases of P3 and P4 duplexes are colored in blue and red, respectively, whereas the backbone of P3 and P4 and all other residues are colored in green

**Table 1 Data collection and refinement statistics**

|  | DNAzyme-Pb$^{2+}$ | DNAzyme | DNAzyme (2′-OMe-G) |
|---|---|---|---|
| *Date collection* |  |  |  |
| Space group | C222$_1$ | C222$_1$ | P4$_3$2$_1$2 |
| Cell parameter |  |  |  |
| *a,b,c* (Å) | 98.7, 118.8, 236.0 | 98.9, 118.9, 236.0 | 78.8, 78.8, 241.7 |
| α,β,γ (°) | 90.0 | 90.0 | 90.0 |
| Wavelength (Å) | 1.0 | 1.0 | 1.0 |
| Resolution range (Å) | 30.0-2.55 | 30.0-3.05 | 30.0-3.80 |
| Outer shell (Å) | 2.64-2.55 | 3.16-3.05 | 3.94-3.80 |
| Completeness (%)$^a$ | 98.6 (95.5) | 97.5 (90.5) | 96.8 (92.9) |
| Redundancy$^a$ | 7.6 (3.6) | 8.3 (4.6) | 8.7 (4.1) |
| *I*/σ(*I*)$^a$ | 26.8 (2.3) | 20.4 (1.9) | 12.2 (1.5) |
| $R_{merge}$ (%)$^a$ | 5.9 (43.2) | 11.0 (35.7) | 6.3 (36.6) |
| *Refinement* |  |  |  |
| Resolution | 2.55 | 3.05 | 3.80 |
| No. reflection | 45,470 | 26,143 | 7857 |
| $R_{work}$ (%)/$R_{free}$ (%) | 23.5/28.3 | 21.5/27.8 | 25.8/28.5 |
| No. of atoms |  |  |  |
| Protein | 5768 | 5742 | 2830 |
| Nucleic acid | 2458 | 2458 | 1208 |
| Pb$^{2+}$ | 1 | 0 | 0 |
| Water | 20 | 0 | 0 |
| B factors |  |  |  |
| Protein | 85.5 | 95.1 | 98.9 |
| Nucleic acid | 91.7 | 91.9 | 99.0 |
| Pb$^{2+}$ | 117.9 |  |  |
| Water | 67.9 |  |  |
| R.m.s. deviations |  |  |  |
| Bond length (Å) | 003 | 0.010 | 0.003 |
| Bond angle (°) | 0.916 | 1.247 | 0.581 |
| Ramachandran plot (%) |  |  |  |
| Most favored | 95.6 | 94.3 | 92.5 |
| Additional allowed | 4.3 | 5.7 | 7.5 |

$^a$Values in parentheses are for the outer shell

provided detailed insights into the Pb$^{2+}$-accelerated catalysis mechanism of 8–17 DNAzyme.

## Results

**Crystallization and structural determination of 8–17 DNAzyme.** In addition to RNAs, 8–17 DNAzyme can also cleave DNA substrates with a single ribonucleotide at the cleavage site (Fig. 1b). To unravel the catalytic mechanism of the DNAzyme, we carried out crystallographic studies of 8–17 DNAzyme. Via screening various sequences, we fortunately obtained the crystals and solved three complex structures (Table 1) of one DNAzyme sequence (Fig. 1b), Dz36, which is 36 nt in length and was designed based on the sequence reported by Santoro and Joyce[3]. Two types of non-cleavable 23-nt substrate analogs, including one native DNA and one DNA with 2′-OMe-G at the cleavage site, were utilized in the structural studies. Dz36 paired with the native DNA substrate in two structures and it paired with the 2′-OMe-G modified DNA in the third structure, which is referred to as DNAzyme(2′-OMe-G) hereafter.

Crystals of one DNAzyme/native DNA complex and DNAzyme-(2′-OMe-G) were grown in the absence of PbCl$_2$, whereas the crystals were grown in the presence of PbCl$_2$ for another DNAzyme/native DNA complex structure, which captured one Pb$^{2+}$ ion and is referred to as DNAzyme-Pb$^{2+}$ hereafter. To facilitate the molecular packing and crystallization, all the crystals were grown in the presence of African swine fever virus DNA polymerase X (*Asfv*PolX), and the structures (Fig. 1c) were solved by molecular replacement method using the *Asfv*PolX structure (PDB ID: 5HRB)[28] as the search model.

**Overall folding of 8–17 DNAzyme.** Among the three structures, DNAzyme-Pb$^{2+}$ complex diffracted to the highest resolution (2.55 Å) and refinement resulted in a well-defined electron density for all the nucleotides (Supplementary Fig. 1a,b). The DNAzyme-Pb$^{2+}$ crystal adopts C222$_1$ space group, it contains two Dz36-substrate complexes per asymmetric unit; each Dz36-substrate complex was clamped by two *Asfv*PolX molecules at the blunt ends (Fig. 1c). In the crystal lattice, no protein–DNA interaction involves the catalytic core region of Dz36. The DNAzyme/native DNA complex without Pb$^{2+}$ adopts the same space group as DNAzyme-Pb$^{2+}$; as indicated by the very low value (0.307 Å) of the root mean square deviations (rmsd), the overall conformations of these two complexes are virtually identical.

Unlike DNAzyme-Pb$^{2+}$, the DNAzyme(2′-OMe-G) structure belongs to the P4$_3$2$_1$2 space group. Owing to the different molecular packing, the relative orientations of *Asfv*PolXs are significantly different in the DNAzyme-Pb$^{2+}$ and DNA-zyme-(2′-OMe-G) structures; however, except the blunt-end base pairs that interact with *Asfv*PolXs, the central regions of the Dz36/substrate complexes are very similar in the two structures (Supplementary Fig. 1c), indicated by the low rmsd value (0.586 Å) between them. These observations suggested that the presence of *Asfv*PolX has very little impact on the folding of the Dz36/substrate complex, especially the core region. Our in vitro assay also suggests that the presence of *Asfv*PolX does not affect the catalytic reaction (Supplementary Fig. 1d).

The Dz36/substrate complexes assemble into a "V" shape with two arms (P1 and P2) orientated ~70° with respect to each other in all the three structures. The two arms are connected through the catalytic core (15 nt) and bind to the substrate forming a dinucleotide kink at the junction (G−1 and G+1; Fig. 1b and d). Unexpectedly, the catalytic core forms a compact small DNA pseudoknot consisting of two short duplexes (P3 and P4) perpendicular to each other (Fig. 1d). P3 is composed of three base pairs (G2:C10, C3:G9, and C4:G8) and it has been previously predicted[3]. The structurally identified P4 contains two base pairs, including one canonical G6:C12 pair and one non-canonical A5:G13 pair (Fig. 1e); these four residues (A5, G6, C12, and G13) are all highly conserved.

**Cleavage site GG-kink.** In all the DNAzyme/substrate complex structures, the substrate cleavage site residues G+1 and G−1 are kinked and pair with the nucleotides T1 and A14 of the DNAzyme, respectively; T1 and A14 also stabilize the arrangement of P3 and P4 (Fig. 2a). T1 packs against P3 and forms a canonical T1:G+1 wobble pair (Fig. 2b), and A14 packs against P4 and forms a non-canonical A14:G−1 base pair (Fig. 2c). In addition to stacking with P3, T1:G+1 pair also stacks with the C−1:G+2 base pair of P1, while A14:G−1 pair primarily stacks with the A5:G13 base pair of P4 (Fig. 2c). The overall fold of the catalytic core is a twisted compact pseudoknot that resembles the shape of an inverted cone (Fig. 2a). On the bottom of the cone, the formed T1:G+1 and A14:G−1 pairs twist G+1 and G−1 away from each other, thereby resulting in a kink (GG-kink) at the cleavage site (Fig. 2a). These intrinsic interactions can stabilize the orientations of G+1 and G−1 and enhance the RNA-cleaving reaction.

**Conformational changes of the catalytic site residues.** The conformations of the DNAzyme-Pb$^{2+}$ and DNAzyme/native DNA complex without Pb$^{2+}$ are virtually identical; in both structures, the N1 atom of G13 forms one hydrogen bond (H-bond, 2.8 Å) with the OP1 atom of G+1 (Supplementary Fig. 2a). These structural similarities indicate that the Pb$^{2+}$-binding site is pre-organized and that the Pb$^{2+}$ binding does

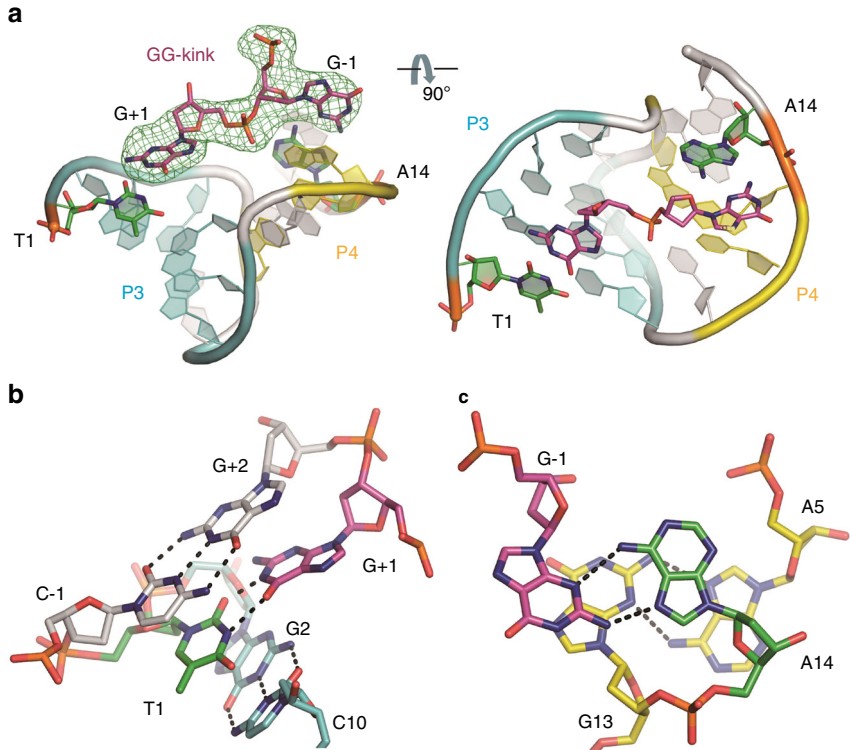

**Fig. 2** Interactions between the catalytic core and the GG-kink. **a** The relative arrangement between the catalytic core and the GG-kink, based on the DNAzyme-Pb²⁺ complex structure. The annealed $F_o$-$F_c$ omit map for the GG-kink is contoured at 3.0 σ level and colored in green. **b** T:G Wobble pair between T1 and G + 1. (**c**) The non-canonical A:G pairing between A14 and G−1. In **b**, **c**, the residues are shown as stick models in the same color scheme for N (blue), O (red), and P (orange) atoms, but the C-atoms are colored in magenta, green, cyan, white, and yellow for G−1 and G+1, T1, and A14, the G2:C10 pair, the last pair of P1, and the A5:G13 pair, respectively

not induce local and global conformational changes of this DNA catalyst. This structural insight is also consistent with the early finding[24] that the additional folding of the DNAzyme is not necessary for the Pb²⁺ cofactor, while Mg²⁺ and Zn²⁺ induces the additional folding of the DNAzyme.

Compared to the real substrate, the G−1 residue of the native DNA lacks one hydroxyl group (O2′) at the 2′ position. To test whether O2′ could affect the local conformation of the catalytic core and to shed more light on the catalytic mechanism of the DNAzyme, 2′-OMe protection was introduced to the cleavage site (rG-1) of the DNAzyme(2′-OMe-G) structure. As aforementioned, the overall folds of the DNAzyme with or without 2′-OMe modification are very similar (Supplementary Fig. 1c); however, structural superposition does reveal some subtle conformational differences at the catalytic sites. As depicted in Supplementary Fig. 2b, although the introduction of the 2′-OMe group did not cause an obvious change to the nucleobase and sugar pucker of G−1 in the DNAzyme-(2′-OMe-G) structure, it led to the shifting of both G13 nucleobase and G+1 phosphate group. When compared to the DNAzyme-Pb²⁺ structure, the distance (4.3 Å) between the N1 atom of G13 and the OP1 atom of G+1 is 1.5 Å longer in the DNAzyme-(2′-OMe-G) structure. The sugar puckers adopt C2′-endo conformations for G-1 residues in both DNAzyme-(2′-OMe-G) and DNAzyme-Pb²⁺ structures, whereas the sugar puckers of G+1 residues adopt the C1′-exo and the C3′-endo conformations in the two structures, respectively.

**General acid–base catalytic mechanism of 8–17 DNAzyme.** Very fortunately, an in-line arrangement of the nucleophile (O2′ atom of G−1), the electrophilic center (P atom of 3′-phosphate of G−1), and the leaving group (O5′ atom of G+1) was captured in the DNAzyme-(2′-OMe-G) structure (Fig. 3a,

Supplementary Fig. 2c); the O2′-P-O5′ angle is 160°. The distance between the N1 atom of G13 and the O2′ atom of G−1 is 3.3 Å, and it is 3.2 Å between the O2′ atom and the 3′-phosphate phosphorus of G−1. The methyl group of the 2′-OMe-G residue is disordered and was not modeled in the structure.

The structural observations (Fig. 3a) suggest that 8–17 DNAzyme may follow an in-line attack mechanism and our structures represent the pre-catalytic state of the RNA cleavage reaction. Besides, these observations also suggest that the nucleobase of the highly conserved G13 residue may play the key role in the catalysis, functioning as the general base to deprotonate the 2′-OH of G-1 for attacking the 3′-phosphate of G−1. To confirm the functional importance of G13, we performed the activity studies using the native and G13-modified DNA-zymes (Fig. 3b, Supplementary Fig. 3). Compared with the native Dz36, methylation at the O6 position (for Dz36-6mG13) lowered the DNAzyme's activity by 40-fold; methylation at the N1 position (for Dz36-1mG13) caused more dramatic reduction on the DNAzyme's activity. These observations suggest that the N1 atom of G13 plays critical role in the catalytic process of the DNAzyme.

In the DNAzyme-Pb²⁺ complex structure, one Pb²⁺ ion was captured in the catalytic core (Fig. 3c). The Pb²⁺ ion coordinates with the O6 atom of G6, which forms G6:C12 pair identified structurally. Interestingly, the Pb²⁺ ion also coordinates with a water molecule interacting with the O5′ atom of G+1. While the distance between the Pb²⁺ and the water is 2.5 Å, the distance between the water and the O5′ atom of G+1 is 3.2 Å, indicating a hydrogen bond formation. This observation suggests that the 8–17 DNAzyme may catalyze the RNA cleavage reaction via the Pb²⁺ indirectly assisted mechanism and the Pb²⁺-coordinated water plays an important role in the catalysis (Fig. 3d). Besides

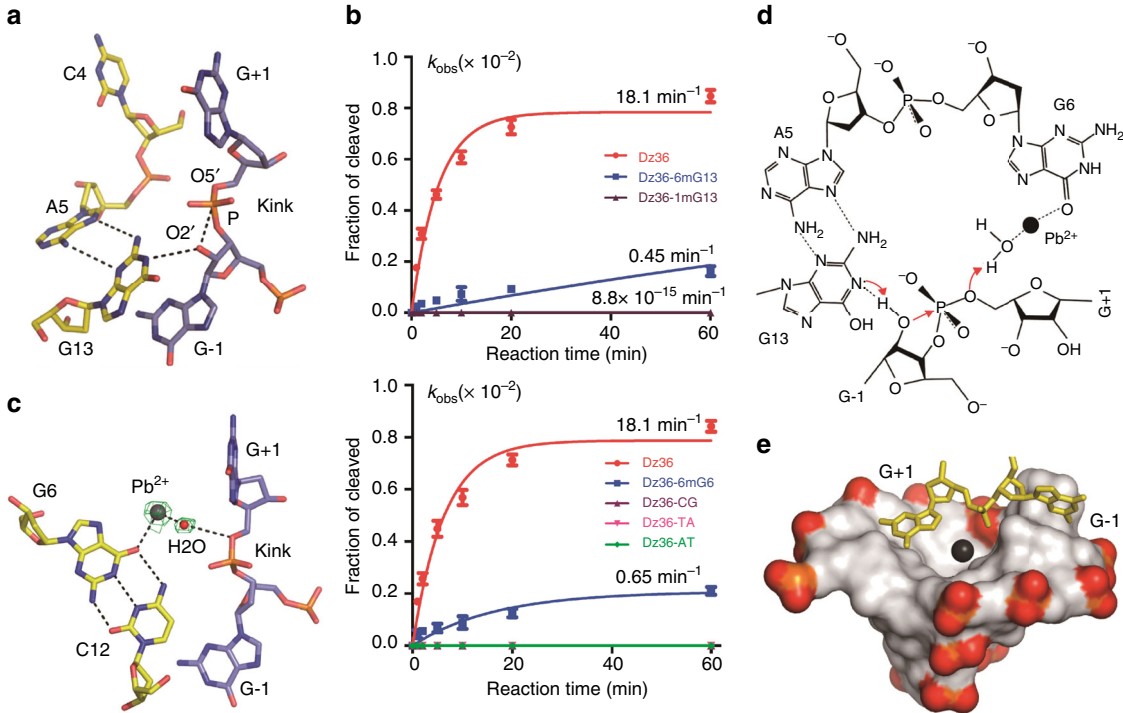

**Fig. 3** Cleavage site assembly and proposed catalytic mechanism. **a** Detailed conformations of the GG-kink and G13 observed in the DNAzyme(2′-OMe-G) structure. **b** Comparison of the catalytic activities of the wild-type DNAzyme, mutants with methylated G6 or G13, and mutants with G6:C12 replaced with other Watson−Crick base pairs. Due to overlapping with Dz36-AT, the curves for Dz36-CG and Dz36-TA are not visible. **c** Coordination of $Pb^{2+}$ and the catalytic water observed in the DNAzyme-$Pb^{2+}$ structure. The annealed $F_o$-$F_c$ omit maps for $Pb^{2+}$ and the water molecule are contoured at 3.0 σ level and colored in green. **d** Proposed mechanism for the RNA substrate cleavage by 8−17 DNAzyme. The N1 atom of G13 deprotonates the 2′-OH group of the attacking G−1 residue. The O6 atom of G6 coordinates with $Pb^{2+}$, which will activate the catalytic water molecule (general acid) to provide a proton to the O5′ atom of the leaving residue G+1. **e** Surface representation showing the preformed cation binding cage. The $Pb^{2+}$ is shown as a black sphere. The G−1 and G+1 residues are shown as stick models in yellow. The phosphorus atoms and the oxygen atoms of the backbone phosphate groups of the DNAzyme are colored in orange and red, respectively

orientating, coordination with the $Pb^{2+}$ ion may also reduce the pKa of the water molecule; this water molecule may then serve as the general acid and provide a proton to the O5′ atom of G+1 for the O5′ leaving, thereby accelerating the RNA cleavage reaction.

Due to the possible quick decay of the crystal and the dynamic binding with the DNAzyme, the occupancy (40%) of $Pb^{2+}$ is not very high in our DNAzyme-$Pb^{2+}$ structure. Although more complex structures with higher resolution are necessary to further verify the identity of $Pb^{2+}$ in the future, the functional importance of G6 and its coordination with the potential $Pb^{2+}$ can be supported by our in vitro activity studies (Supplementary Fig. 3). As depicted in Fig. 3b, methylation at the O6 position (for Dz36−6mG6) lowered the DNAzyme's activity by about 30 folds, replacing of the G6:C12 pair with any other Watson−Crick base pairs (such as C6:G12 in Dz36-CG, T6:A12 in Dz36-TA, and A6: T12 in Dz36-AT, Supplementary Table 1) almost completely abolished the DNAzyme's activity.

The $Pb^{2+}$-binding cage is located between the GG-kink and the center of the catalytic core (Fig. 3e). Although it only coordinates with the O6 atom of G6 and the catalytic water molecule, the distances between $Pb^{2+}$ and the heteroatoms of the surrounding residues are mainly within the range of 5−7 Å (Supplementary Figs. 4a, b), suggesting that this cage could accommodate one multi-coordinated $Pb^{2+}$ ion. Besides water molecules, metal cations also frequently coordinate with negatively charged ligands, such as the phosphate groups of DNAs or RNAs; however, as depicted in Fig. 3e and Supplementary Fig. 4c, all the phosphate groups of the catalytic core residues are located at the

edge and are not suitable for coordinating with a cation bound at the cage. All together, these observations suggest that, instead of the space availability, a lack of an electrostatic interaction might be the main reason that $Mg^{2+}$ and other cations are excluded from the $Pb^{2+}$-binding cage; $Mg^{2+}$ and some cations could bind to alternative sites of the DNAzyme, which will cause additional folding of the DNAzyme and lead to the lower catalytic activity[24].

**Characterization of the catalytic core site residues**. On the basis of our structural analysis, we decided to further investigate a few other key nucleotides the mutations and activity measurements, such as C7, T11, A14, and A15. According to the crystal structure, C7 and T11 are located at the two junction sites (labeled as J3/4 and J4/3) and stack with C4:G8 pair of P3 and G6:C12 pair of P4, respectively (Fig. 4a−c), thereby stabilizing these two short duplexes of the pseudoknot. Previous studies showed that mutations of C7 and T11 had no obvious impact on the activity of this DNAzyme[29]. However, our in vitro cleavage assays (Fig. 4d, Supplementary Fig. 5, and Supplementary Table 1) showed that deletion of C7 (for Dz36-del7) at the J3/4 site lowered the DNAzyme's activity by about 100-fold; although it is not as dramatic as the C7 deletion, deletion of T11 at the J4/3 site also lowered the DNAzyme's activity by about 20-fold. C7 and T11 neither pair with other residues nor are directly involved in catalysis, the mutation and structure studies indicate that they may play important roles structurally.

8−17 DNAzyme has two types of consensus sequences (Fig. 1a), one is shorter than another by one nucleotide at position 15,

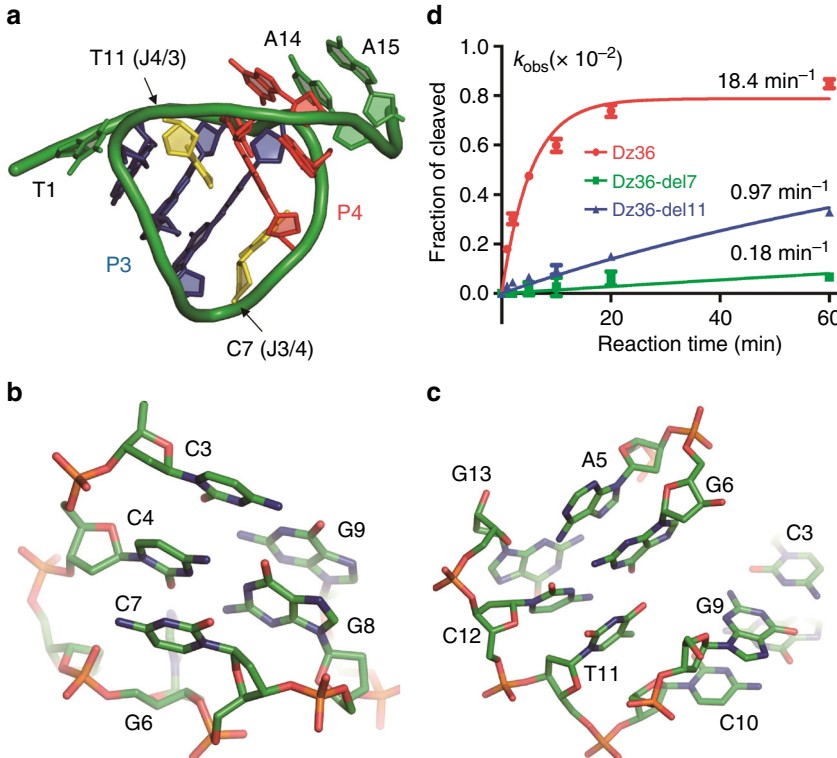

**Fig. 4** Functional characterization of the bridging residues. **a** Cartoon representation showing the overall fold of the catalytic core. The base pairs of P3 and P4 are colored in blue and red, respectively. The bridging residues C7 and T11 are colored in yellow. Stick representation of **b** C7 at the bridging site J3/4 and **c** T11 at the J4/3 bridging site. **d** Comparison of the catalytic activities of the wild-type DNAzyme and mutants with C7 or T11 deletion

corresponding to A15 of Dz36. As revealed by the crystal structure (Fig. 5a), A15 is sandwiched between the A16:T-2 pair (the first base pair of P2 duplex) and the A14:G-1 pair. To further clarify the functional role of A15, we constructed one mutant with A15 deleted (for Dz35) and carried out in vitro cleavage assays (Fig. 5b, Supplementary Fig. 6, and Supplementary Table 1). Although it is not very different on the substrates with purines at the cleavage site (SrA and SrG), the cleavage activity of Dz35 is about 5-fold lower than the native DNAzyme (Dz36) toward the substrates with pyrimidines at the cleavage site (SrC and SrU). These observations suggest the DNAzyme can tolerate the deletion of A15, whereas presence of A15 does have some enhancing effect on DNAzyme's activity.

In the DNAzyme structure, A14 and G-1 forms one non-canonical A:G pair (Figs. 2c and 5a). To verify the functional importance of this non-canonical base pairing, we further replaced the A14 residue of Dz35 by dG (for Dz35-G), dC (for Dz35-C), or dT (for Dz35-T) and carried out in vitro cleavage assays (Fig. 5b, Supplementary Fig. 6, and Supplementary Table 1). Similar to Dz35, the overall pyrimidine substrate (SrC and SrU) cleavage activities of the three Dz35 mutants are significantly weaker than their purine substrate (SrA and SrG) cleavage activities. Compared to Dz35, the SrG cleavage activities of Dz35-G and Dz35-T are 2-fold lower, and it is about 6-fold lower for the Dz35-C mutant; the lowest activity of Dz35-C might be caused by the Watson–Crick C14:G-1 pairing between the DNAzyme and the substrate. For the SrA substrate, the cleavage activity of Dz35-G is the weakest, it is only about 20% of Dz35 and about 40% of Dz35-C and Dz35-T. Similar to previous observations[3], our results further indicate that 8–17 DNAzyme prefers the purine substrates; although the T-rA combination can be well tolerated, the DNAzyme normally prefers a non-

Watson–Crick paired combination (especially the A-rG pair) at the catalytic site.

**Comparison with the RNA-cleaving ribozymes.** Similar to 8–17 and 10–23 DNAzymes, some ribozymes can also catalyze the RNA cleavage reaction; the structures of these RNA-cleaving ribozymes have been well-characterized. The size of the catalytic core and the overall fold of 8–17 DNAzyme are very different from these RNA-cleaving ribozymes, including leadzyme[30,31], HDV ribozyme[21,22], and hairpin[20] ribozyme (Supplementary Fig. 7a–d). Unlike the V-shape of 8–17 DNAzyme, the leadzyme adopts a pseudo duplex-like conformation with the cleavage site residues bulged out; the hairpin ribozyme has a four-way junction-like fold. Interestingly, although the conformations of the enzyme strands are significantly different in the 8–17 DNAzyme and the hammerhead ribozyme[19] structures, their substrate strands adopt very similar conformations, especially at the kinks and the flanking regions (Supplementary Fig. 7e–f).

The cleavage activities of the leadzyme and HDV ribozyme are $Pb^{2+}$ and $Mg^{2+}$ dependent, respectively. As indicated by the leadzyme-$Sr^{2+}$ and HDV-$Mg^{2+}$ complex structures, the cation cofactors coordinate with the phosphate groups of the enzymes (Fig. 6a, b); however, may due to the unique folding, $Pb^{2+}$ ion does not coordinate with any phosphate group within the catalytic core of 8–17 DNAzyme. These structural observations further suggest that $Pb^{2+}$ is very flexible in coordination. Among the common cations, $Ba^{2+}$ is the one that can most closely mimic $Pb^{2+}$ in coordination; however, previous studies showed that the $Ba^{2+}$-assisted RNA cleavage reaction of 8–17 DNAzyme is much slower than the one supported by $Pb^{2+}$. Compared to $Ba^{2+}$ and many other cations, such as $Mg^{2+}$, $Mn^{2+}$, $Ca^{2+}$, and $Co^{2+}$, the pKa value of hydrated $Pb^{2+}$ is much lower[18]. Although it is not

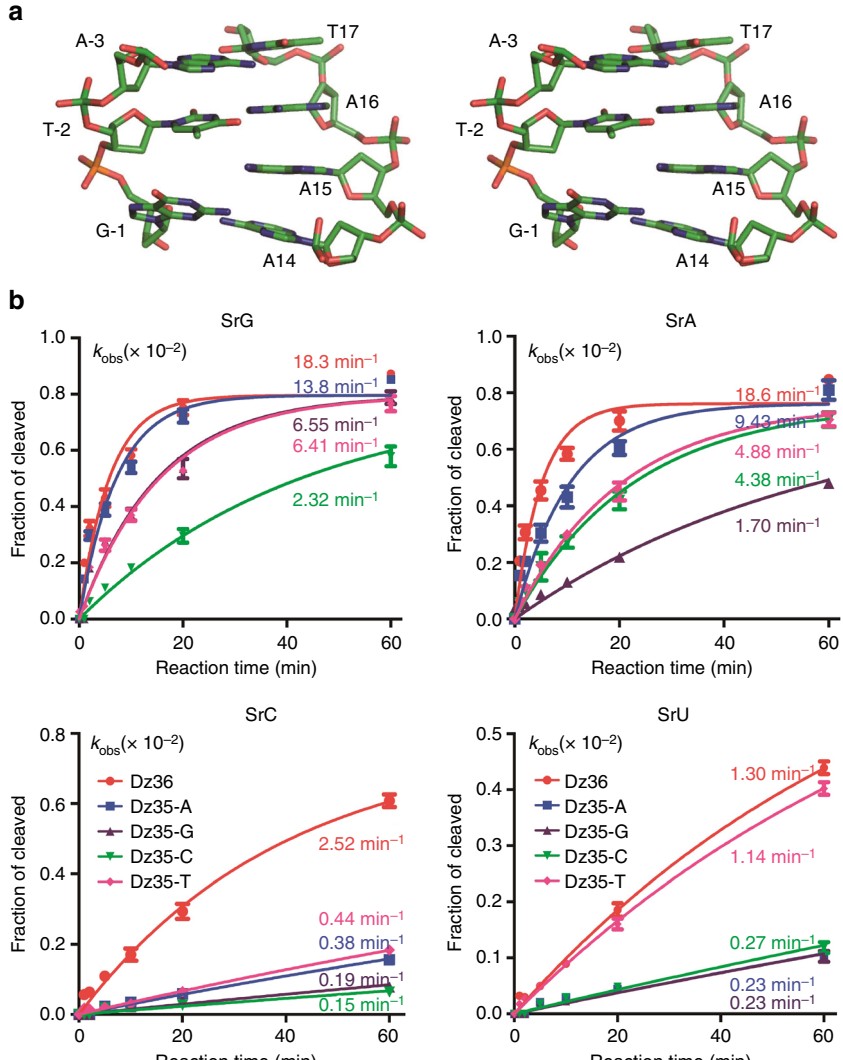

**Fig. 5** Functional characterization of A14 and A15. **a** Stereo-view showing the detailed conformation of A14 and A15. **b** Comparison of the catalytic activities of the wild-type DNAzyme, mutant with A15 deletion, and mutants with both A15 deletion and A14 mutation. Four different DNA substrates with single ribonucleotide at the −1 position were utilized in the catalysis. The color schemes of the DNAzymes are identical in all the panels in **b**

strictly correlated to the reverse order of pKa, the low pKa of hydrated $Pb^{2+}$ may contribute to the catalytic efficiency of 8–17 DNAzyme. Coordination with $Pb^{2+}$ ion activates the catalytic water molecule, which will serve as general acid and provide a proton for displacing O5′ atom of G+1 (Fig. 3d).

In the proposed mechanism of the 8–17 DNAzyme (Fig. 3d), the nucleobase of the highly conserved G13 serves as the general base, which deprotonates 2′-OH of G−1 for in-line attacking the 5′-phosphate phosphorus and breaking the P-O5′ bond of G+1. Although it was not observed in the leadzyme-$Sr^{2+}$ and HDV-$Mg^{2+}$ complex structures, which were all captured at the inactive ground states (Fig. 6a, b), the in-line arrangement was observed in the hairpin and hammerhead ribozyme structures (Fig. 6c, d). These observations suggest that the in-line attacking mechanism supported by the general acid and base might be a common catalytic mechanism for the RNA-cleaving DNAzymes and ribozymes. Similar to the 8–17 DNAzyme structure, the general bases are also played by G nucleotides in the hammerhead and hairpin structures, which are G36 and G8, respectively; during the catalysis, these catalytic G residues will deprotonate the 2′-OH at the cleavage site.

Interestingly, further structural comparison revealed some other similarities between 8–17 DNAzyme and the RNA-cleaving ribozymes, especially the hammerhead ribozyme. As depicted in Fig. 6e, the conformations of the sugar puckers of the −1 site resides and the backbone phosphate groups of the +1 site residues are almost identical in the two structures, which further supports the conserved in-line attacking mechanism. Although the orientations of the sugar puckers and the nucleobases are different for the +1 site resides, the relative kinking angles between −1 and +1 site residues are comparable, leading to the similar folding of the substrate strands in the two structures (Supplementary Fig. 7f). Similar to the G13:A5 pair of the 8–17 DNAzyme structure, G36 of the hammerhead ribozyme also forms one non-canonical G:A pair with residue A21 (Fig. 6d); in the hairpin ribozyme structure, the conserved G8 interacts with U2 (Fig. 6c). Interactions between non-Watson–Crick paired residues are also observed for the −1 site residues, such as A14:G-1 (Fig. 2c), A9:A-1 (Fig. 6c), and A37:C6 (Fig. 6d) in the 8–17 DNAzyme, hairpin and hammerhead ribozymes, respectively. Compared to the Watson–Crick-paired residues, interactions between these non-Watson–Crick paired residues are more

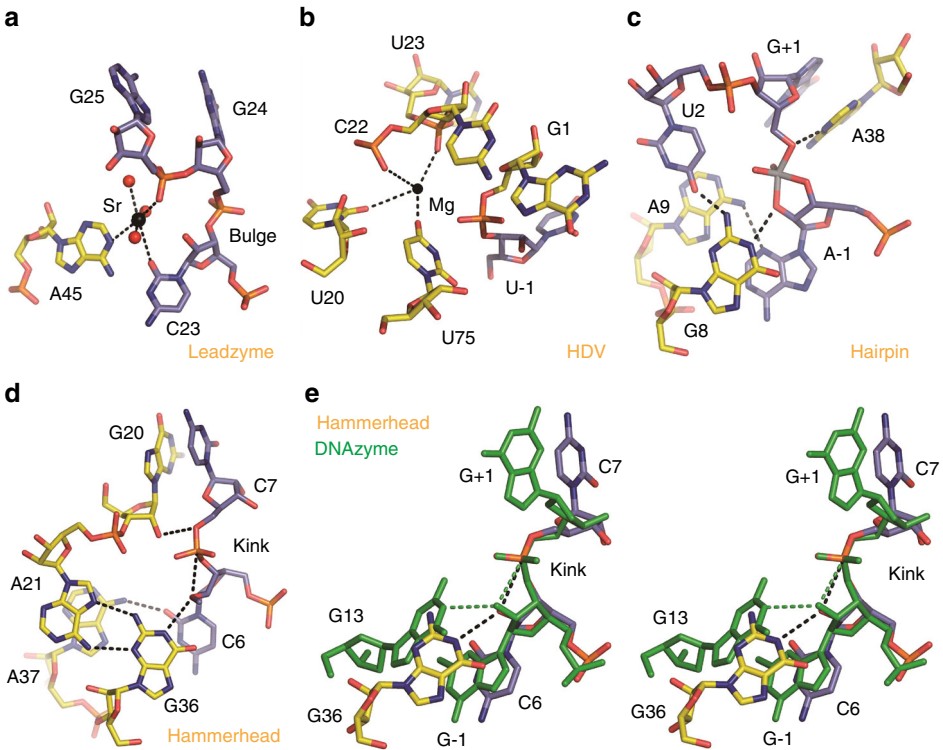

**Fig. 6** Comparison of the catalytic site architectures between 8–17 DNAzyme and RNA-cleaving ribozymes. The catalytic site architectures observed in the **a** leadzyme (PDB_ID:1NUV), **b** HDV (PDB_ID:1JS3), **c** hairpin (PDB_ID: 1M5O), and **d** hammerhead (PDB_ID: 3ZP8) ribozymes. **e** Superposition of the catalytic site residues in the hammerhead ribozyme and 8–17 DNAzyme structures. 8–17 DNAzyme structure is colored in green in **e**. In **a**–**e**, the C-atoms are colored in yellow and light blue for ribozymes and substrates, respectively, whereas other atoms are colored with the same scheme (N, blue; O, red; P, orange) for both ribozymes and substrates. $Sr^{2+}$ ion, water molecules, and $Mg^{2+}$ ion captured in the leadzyme and HDV ribozyme structures are shown as spheres in black, red, and black, respectively

dynamic and flexible, which may provide the structural basis for the conformational changes associated with the catalytic process.

## Discussion

Here, we present the structural and functional studies of 8–17 DNAzyme. Our structures capture the pre-catalytic state of the RNA cleavage and highlight the unanticipated folding, compacted pseudoknot, $Pb^{2+}$ catalysis, and 3D interactions that allow DNA to fold and function as a catalyst. We found that the 8–17 DNAzyme adopts a V-shape fold consisting of two helical substrate-recognizing arms and one twisted DNA pseudoknot, that the DNAzyme forces RNA substrate to bend and form a 2-nt-kink at the cleavage site, and that many residues in the 15-nt catalytic core are essential for the DNAzyme activity. Our crystal structures have highlighted that similar to its counterpart ribozymes, this DNAzyme catalyzes the RNA cleavage via a general acid-base mechanism. Our structures represent the catalytic form RNA-cleaving DNAzyme structure; in combination with the RNA-ligation DNAzyme structure (the only reported catalytic form DNAzyme structure), our studies further indicate that the DNA sequences, with flexible backbone, sugar pucker and nucleobases, are capable of offering diverse structures and functions. Clearly, the biopolymers can provide a novel solution for a functional need in catalysis.

## Methods

**Oligonucleotide synthesis**. All the unmodified DNAzymes (Supplementary Table 1) were ordered from the Shanghai GENERAY Biotech Co., Ltd, China (http://www.geneary.com.cn) and purified by HPLC in the laboratory. The

methylated DNAzymes (Dz36–6mG6, Dz36-1mG13, and Dz36-6mG13) were purchased from the Bioneer company (http://www.bioneer.com). The substrates (Supplementary Table 1) with single ribonucleotide at the cleavage site (N-1 site) or 2′-OMe-G methylation modification were synthesized in the laboratory using the 394 DNA/RNA synthesizer (Applied Biosystems) and purified by both HPLC and denaturing urea PAGE gel. All the oligonucleotides were dissolved in RNase-free ddH$_2$O and the concentrations were measured using a UV-spectrometer.

**In vitro cleavage assay**. 5′-FAM labeled substrates were ordered from Sangon Biotech and further purified by 8 M urea gel purification in the laboratory. A 6 μl reaction system comprising 3 μl 2× reaction buffer (800 mM KCl, 200 mM NaCl, 100 mM Hepes-NaOH pH 7.5, 0.5 mM PbCl$_2$), 1.5 μl enzyme (0.2 μM) and 1.5 μl substrates (0.4 μM) were mixed separately. Reactions were incubated at 26 °C and quenched by 6 μl Formamide Loading Buffer (90% formamide, 40 mM EDTA, 0.01% xylene cyanol) at various time points. Samples were heated at 95 °C for 3 min, centrifuged and loaded onto 8 M urea 18% PAGE. Gels were run at 10 W for 60 min in 0.5× TBE buffer and scaned by TyphoonTM FLA 9000 Imaging Scanner.

Time courses for each reaction condition were repeated for at least three times. The cleaved products and uncleaved substrate were quantified with ImageQuant TL. Data were then fitted to the exponential $Y = Y_{max}[1-e^{(-K_{obs}t)}]$ using non-linear regression in GraphPad Prism 5. The observed rate constant ($K_{obs}$) and maximum cleavage yield ($Y_{max}$) were determined from the regression curve.

**Crystallization and data collection**. Crystallization samples were formed by mixing the DNAzyme (in ddH$_2$O), substrate analog DNA (in ddH$_2$O), AsfvPolX protein (in 50 mM Tris, pH 7.5, and 300 mM NaCl) and PbCl$_2$ without pre-annealing. The AsfvPolX protein was expressed and purified in the laboratory as previously described. The final concentrations of the DNAzyme, substrate DNA analog, and AsfvPolX were 0.3, 0.3, and 0.25 mM, respectively; the PbCl$_2$ concentration was 1.0 mM if present. The initial crystallization conditions were identified at 16 °C using the Gryphon crystallization robot system (Arts Robbin Instruments) and commercial crystallization kits (Hampton Research). The sitting-drop vapor diffusion method with the 3-drop intelliplate plates were utilized for initial screening and optimization performed at 18 °C. The drop contains 1 μl sample and 1 μl crystallization buffer, which is composed of 0.2 M di-ammonium

hydrogen phosphate and 20% PEG 3350 for the DNAzyme and DNAzyme-Pb$^{2+}$ crystals, and is composed of 0.1 M Tris (pH 7.0) and 15% v/v ethanol for the DNAzyme-(2'-OMe-G) crystals. All crystals grew in 2 days and reached their full sizes in a week.

The crystals were cryoprotected by sequential soaking in their mother liquid supplemented with glycerol (from 5 to 20%) and flash-frozen by quickly dipping into liquid nitrogen. The X-ray diffraction data were collected at beamlines BL17U and BL19U of the Shanghai Synchrotron Radiation Facility (SSRF) at cryogenic temperature, maintained with a Cryogenic system. One single crystal was used for all data sets; data processing was carried out using HKL2000 or HKL3000[32] programs. Data collection and processing statistics are summarized in Table 1.

**Structure determination and refinement**. All DNAzyme structures were solved by molecular replacement (MR) using the Phaser program[33,34] embedded in the CCP4i suite. The *Asfv*PolX structure was used as the search model. The electron density map revealed the DNAzyme and the substrate analog, which were manually built using COOT[35]. The methyl group of 2'-OMe-G is disordered, therefore it was not modeled in the model. The model was then refined against the diffraction data using the Refmac5 program[36] of CCP4i and phenix.refine program[37] of Phenix. During refinement, 5% randomly selected data was set aside for free R-factor (cross validation) calculations. The 2F$_o$–F$_c$ and F$_o$–F$_c$ electron density maps were regularly calculated and used as guide for model building and adjustment. The phosphate ions, H$_2$O molecules, and Pb$^{2+}$ ion were added at later stage of refinement on the basis of difference electron density. The structural refinement statistics are also summarized in Table 1.

**Data availability**. Structure factors and coordinates have been deposited in the Protein Data Bank under accession codes 5XM8, 5XM9, and 5XMA for the DNAzyme-Pb$^{2+}$ complex, DNAzyme, and DNAzyme-(2'-OMe-G) structures, respectively. The data that support the findings of this study are available from the corresponding author upon request.

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

## Acknowledgements

We thank Professor Jack W. Szostak at Harvard University, Professor Dinshaw J. Patel at Memorial Sloan-Kettering Cancer Center, and Professor Xinhua Ji at the NIH for helpful reading and insightful discussions. All X-ray data were collected at Shanghai Synchrotron Radiation Facility beamlines BL17U and BL19U. This work was supported by the Key Research and Development Project of China (2016YFA0500600), the National Natural Science Foundation of China (31370728, 31230041, 21572146), the National Basic Research Program and Sichuan S&T Programs of China (2011CB966304 and 2012CB910502, 2016HH0011), and the US NIH (R01GM095881, R42ES026935).

## Author contributions

H.L., L.Z., and J.G. determined the crystal structure. X.Y. performed the in vitro catalytic assays. H.L., P.H., and R.W. synthesized the nucleic acids. Y.C. and J.Z. purified the

protein. Y.C., J.Z., and B.W. collected the X-ray diffraction data. H.L., S.L., J.Lin, J.Li, J.S., Z.H., J.M., and J.G. designed the study, analyzed the data, and wrote the manuscript.

## Additional information

**Competing interests:** The authors declare no competing financial interests.

