## [Peer Review File · Nature Communications]

Reviewers' comments:

Reviewer #1 (Remarks to the Author):

The study by Liu and coworkers is focused on determining the crystal structure and elucidating the catalytic mechanism of the RNA cleaving DNAzyme “8-17”. The authors determined the 2.55 Å X-ray crystal structure of a sibling of the famous 10-23 DNAzyme and showed how a small DNA motif forms an amazing functional unit. This structure, to my knowledge, is the first structure of an RNA-cleaving DNAzyme. Most importantly, these DNAzymes are not some kind of oddity but are important tools used, for example, for fragmentation of natural mRNAs for subsequent analysis (see a recent example in Luciano et al, 2017, Mol. Cell). They also have potential for silencing harmful genes in mammals.

The authors are very fortunate to obtain the structure in the conformation that appears to reflect a relevant pre-catalytic state. The structure, structure-guided mutagenesis and nucleotide substitution analysis provide a plausible mechanism for catalysis and explain the role of various parts of the DNA in the structure formation and catalysis. In addition, the structure visualized the bound cofactor, a lead cation. I find this study of great interest to general audience of Nature Communications and strongly suggest considering the manuscript for publication after a major revision.

The manuscript has some deficiencies, which have to be addressed prior to publication. The main critique is that the functional assays are not quantitative. Although the gels showing cleavage with different mutations and substitutions provide nice side-by-side comparisons of DNAzyme activity, I would expect the 21st century mechanistic study to rely on measured rate constants and not on qualitative descriptions such as “nearly abolished the enzymatic activity”, “almost completely abolished the activity”, “note as dramatic as”, “significantly reduce the activity”, etc. Based on these gels, I disagree with some of authors’ conclusions, and determination of rate constant values will resolve the controversy. Since the authors already have materials and the assay in hand, these experiments should take a couple of weeks.

I also do not understand the structural basis for the lead cation selectivity and would feel more comfortable about the proposed mechanism of catalysis with less disruptive nucleotide substitutions (probably hard to do) and analysis of the published biochemical data based on structural observations. The authors also did not specify clearly that their structure is all-DNA, and not a DNAzyme bound to the RNA substrate, and that the structure represents one of the variants, albeit the most efficient, of the 8-17 DNAzyme. Overall, the manuscript would benefit from more clarity and comparison with a relevant structure of the leadzyme and not distinct structures of some ribozymes.

Specific critiques and suggestions to improve the manuscript are following.

1. Line 41-42. Please cite the original publications and not follow-up studies in this context. For example, refs. 2-5 should be replaced by the original findings, refs. 13, 21, 22, etc.
2. Line 46-48. In addition to the potential use for silencing genes *in vivo*, RNA cleaving DNAzymes, including 10-23 enzyme, have other applications (see my comment earlier).
3. The authors should carefully review all statements referring to prior studies to avoid factual errors. I have immediately noticed a few and suspect that there are more erroneous statements. For example, in lines 64-66, they cite ref. 21 published in 1997 and then say “later it turned out” and cite ref. 22, which was published in 1996.
4. Line 54. I would avoid citing ref. 22 about Mg5 enzyme and some other papers here and cite the original publication on the 8-17 DNAzyme.
5. Line 63. Please provide the consensus sequences for the motif from the SELEX experiments in Fig. 1A. The crystallized variant belongs to one of the two consensus motifs described by Santoro and Joyce.
6. Line 66. The fact that the 8-17 DNA works better with transition metal cations is not surprising. There is a chemical explanation for this fact.
7. Lines 83-84. Please spell out what was crystallized: a 36-mer DNA paired with a 23-mer substrate DNA (not RNA!) in the absence and presence of the cleavage cofactor, lead cation, as well as a DNA paired to a modified non-cleavable substrate that mimics the RNA substrate.
8. Line 85. Here, the overall structure could be first presented. I recommend showing the entire structure in Fig. 1. An image similar to SFig. 1a would be most useful. Please show nucleobases with sticks to simplify identification of nucleobases. Presentation in Fig. 1d does not let distinguish different purines. To simplify the view, sugars could be reduced to sticks. This image should be color-coded as in Fig. 1 b, with both P3 and P4 is respected colors. The authors may retain green color for the cartoon showing the backbone of the catalytic core but have nucleobases in cyan and green.
9. Fig. 1c. The refined 2Fo-Fc electron density map does not illustrate the quality of the map. Please show the structure with an unbiased map (simulated annealing omit map) and move the figure to the Supplementary Figures section. Please retain the unbiased electron density map for the GG kink in the main text figure.
10. Fig. 1d. With a new image of the structure, this panel might show zoomed-in view of the core, with less of P1 and P2. In addition, a “side” view (rotated 90 degrees along “y axis”) from the “P2 side” could help to understand the formation of the P3 “helix”. Please label P3 and P4.
11. Fig. 1d. The authors highlight C7 and T11 here without explanations. The explanations are given in the second part of the write-up, which however does not refer to this figure.
12. Fig. 1e. This view is difficult to understand because of the oversimplified presentation of the majority of nucleobases, shading of sugar rings, and insufficient transparency of overlapped nucleobases. Please add more labels for nucleotides.
13. Fig.1 legend. “Globe” or “global” architecture?
14. Lines 92-93. I would expect a larger (closer to 180 degrees) angle between the arms

according to smFRET measurements of the Pb²⁺-induced cleavage (Kim et al, 2007, NCB, 3: 763). Is crystal packing affect the angle? Can the angle get smaller without disruption of the structure? Please discuss this issue.

15. Lines 98-100. In Fig. 1, G13 and A5 look like they are not in the same plane while G13 seems to be in the same plane with C12 and G6.

16. Line 97. C4:G8.

17. Line 106. What does “non-canonical T1:G+1 wobble pair” mean? This is a “canonical” wobble base pair.

18. Fig. 2a. This panel shows a surface view which was not discussed in the manuscript.

19. Fig. 2b. Please label interacting nucleotides.

20. Line 109. ... pseudoknot that resembles the shape of an inverted cone.

21. Fig. 3 could be moved to the Supplementary Figures section. Please provide rmsd for the cores of the superposed structures.

22. Line 123. Please provide ref(s) for “early finding”.

23. Line 126-128. This section is misleading. It should be clearly explained that, first, the authors have crystallized the DNAzyme bound to a non-hydrolysable substrate strand made of DNA. Second, they wanted to introduce a ribonucleotide into the DNA strand to obtain a hydrolysable substrate but instead they incorporated a methyl-2'O substituted nucleotide to prevent possible cleavage. The structure was however obtained without Pb, therefore no cleavage was expected. I am wondering why the authors did not obtain the structure with a ribonucleotide in the absence of lead or the structure of the methylated substrate with lead. The structure of the methylated RNA substrate with lead would most closely represent a pre-catalytic state of the enzyme, but this structure was not obtained for unclear reasons. Right now, the catalytic mechanism model represents a combination of information from three structures and none of the structures individually corresponds to a pre-catalytic state structure.

24. Lines 128-130. There must be changes in the conformation of the GG kink, at least in the sugar-phosphate backbone. Please describe these changes, with emphasis on sugar conformations, and show a zoomed-in view of the differences.

25. Lines 130-134 and Fig. 4A. It is very hard to understand from Fig. 4a whether the alignment is indeed in-line or similar to in-line, as was reported earlier in the leadzyme ribozyme structure. Please make a better presentation of this important part, label the atoms involved in the alignment, show the near 180 degree angle (what is the angle value?), and show a zoomed-in view of the unbiased electron density map for the G-1 sugar and the adjacent atoms of the backbone to prove the alignment. Does the G-1 sugar adopt the C3'endo conformation?

26. Line 133. Please split the sentence. Atoms of G13 are not involved in the in-line alignment.

27. Line 135-142. The RNA field experienced many issues when relating ribozyme structures and catalytic mechanisms. Please use very careful wording discussing the catalytic mechanism of the 8-17 DNAzyme. Addition of a methyl moiety to the Watson-Crick edge of G13 can disrupt the structure and indirectly affect catalysis. This is what probably happens with the Dz36-6mG13 enzyme.

28. Lines 144-150, 158-169, Fig. 4c,e. Please show the Pb^{2+} binding site in more detail, with distances to heteroatoms of RNA. It is hard to imagine that a specifically bound cation nicely sitting in the cavity forms only a single coordination bond and nothing else participates in the cation binding. Can Pb^{2+} cation coordinate to N7 of G6? Other metal cations were found interacting with N7 of purines.
29. Please show a lead cation in “real size” (~ 3.0 Å diameter) in Fig. 4 to help better understand space requirements.
30. Line 145. Why is coordination of a Pb^{2+} cation to water unexpected? Metal-water coordination is often involved in catalytic reactions.
31. Line 148-150. Please tone down this statement. The water molecule might be critical; however this conclusion is based only on a sole H-bond distance.
32. Lines 151-156. Again, as in the earlier comment, these nucleotide changes may disrupt the structure and, without experimentally determined structures for mutant DNAzymes, conclusions must be carefully worded.
33. Lines 163-165. The reasons for higher catalytic activity in the presence of lead is not clear. Does a lead cation bind the enzyme better than other cations or is it better at the chemical step?
34. Does comparison of the DNAzyme activities with different cations correspond to the reverse order of pKa of the corresponding metal hydrates? What’s a pH dependence of the cleavage rate with lead? Please discuss prior publications focused on the reaction chemistry in the context of the structure. Can all published biochemical data be explained by the structure-based catalytic mechanism?
35. Lines 163-165. I do not follow authors’ idea and I do not see any structural reason for a lead cation to be a specific binder. Do the authors mean that a shallow cavity is too big for smaller than Pb^{2+} cations or too small for water-coordinated cations (such as fully hydrated Mg^{2+} cation) so that they cannot form a “productive” contact with an active water molecule and bind to the other “sides” of the cavity, away from the catalytic site? There are many precedents when cations such as Mg^{2+} or K^+ bind nucleic acids without coordinated water molecules.
36. It is surprising that, having the crystals without bound cations in hand, the authors did not soak other cations, such as Mg^{2+} or its analogs with stronger anomalous scattering properties (Mn^{2+} , Ca^{2+}), to map the binding sites for these cations and put the issue to rest.
37. Does Co hexamine, a mimetic of a fully hydrated Mg^{2+} cation, support the reaction?
38. The related DNAzyme 17E (Li, 2000, NAR, 28: 481-488) shows reduced activity in the presence of 150 mM F^- anions, consistent with the possibility that a fluoride can replace an active water molecule (as shown for some protein enzymes); however the reaction is not abolished, arguing against a water molecule playing a critical role in the catalysis. Please comment on this observation.
39. The proposed catalytic mechanism requires deprotonation of N1 of G13. This has not been discussed. What would shift the pKa of this group?
40. Is there anything to stabilize the transition state?
41. Does Ba^{2+} cation, the most close mimetic of Pb^{2+} cation, support the reaction?

42. What's the occupancy of Pb²⁺ in the structure?
43. Please show an omit and anomalous maps for Pb²⁺ cation. The authors must prove the identity of lead because the lead-containing structure was determined at higher resolution than the other structures and it has 20 water molecules emerged because of improvement of resolution.
44. Please use the same color for lead cation throughout the figures. In Fig. 4, lead shown in black, grey and green colors, with labels in black and green.
45. Crystallographic table does not list B-factors for DNA, metal and water molecules.
46. Fig. 4e. A standard blue-red presentation of the surface potential could be a better option for this panel.
47. Line 483. Why is the water molecule shown in cyan and not in standard red color for an oxygen atom, with density in green or blue? What's the B-factor of this water molecule?
48. Fig. 4d. G+1 is a deoxyribonucleotide and therefore it should not have a 2'-OH group unless the authors say in the figure legend that they are showing all-RNA substrate.
49. Fig. 4d. Why are labels shown in two colors?
50. Please compare the structure and the putative catalytic mechanism of the DNAzyme with the structure and catalysis of the leadzyme.
51. Line 173. ...activity measurements...
52. Fig. 5a. This figure is very crowded and unclear. See my earlier comments for Fig. 1e to improve the view.
53. Fig. 5a. Is T11 cyan or dark blue? What's the magenta nucleotide?
54. Fig. 5b,c. Motivation for showing the electron density map is not clear. These are not the most important regions of the DNAzyme and the refined 2Fo-Fc map is not the best way to illustrate the quality of the structure.
55. Lines 177-180 and 182-184. Motivation for testing an insertion of nucleotides at position 7 (4 mutants in total) and conclusions are not obvious. What do these mutations address? C7 provides spacing between adjacent base pairs and the structure shows that there is enough space for looping out a residue without impact on catalysis unless the inserted base is capable of forming alternative base pairs and disrupting the fold. That's what the authors see: insertion of purines that have better potential for base pairing is more disruptive than insertion of pyrimidines.
56. Lines 180-182. This is an incorrect conclusion. While deletion of T11 does significantly reduce cleavage, insertions do not affect cleavage efficiency strongly, leading to the same conclusion as before: an insertion can be tolerated with only small impact on activity.
57. Line 185. Presented figures do not help to evaluate the potential role of A15 and A14.
58. Lines 185-187. If I understand correctly authors' idea, deletion of A15 converts the 5'-WCGAA consensus sequence into the 5'-WCGR sequence, both observed in the original publication (Santoro et al). This result means that both consensus sequences from SELEX correspond to the same DNAzyme structure.
59. Line 189. There is no Fig. 3c in the paper.
60. Lines 188-190. This conclusion is not entirely correct. According to the original observation

(Santoro et al) and SFig. 3A, the A14G substitution in the context of the delA15 shows some activity. Same is true for A14T. This means that the A14:G-1 pair is important but not critical for catalysis.

61. Lines 190-193. This sentence is also not entirely accurate. While the majority of Watson-Crick pairs replacing the A14-G-1 base pair show diminished activity, the T-rA combination is rather active (SFig. 3b) and several non-canonical base pairs, A-rA (SFig. 3a), A-rC (SFig. 3d), T-rC (SFig. 3d) and T-rU (SFig. 3s) also show good activity. The authors cannot make strong conclusions without measured rate constants. It is recommended to provide a supplementary figure with a structure-based schematic of these combinations and discuss similar data from prior publications.

62. Lines 193-195. The kink is usually stabilized through extensive stacking interactions and that's what the structure shows. The identity of base pairs, Watson-Crick or non-canonical, for making a bent in the backbone should not matter. What is likely important is that non-canonical base pairs are more dynamic than canonical base pairs and therefore offer flexibility required for the catalytic reaction. Published articles have discussed this point and the authors may discuss it from the structural perspective as well.

63. Line 195. Please provide canonical designations of the South (C2'-endo) and North (C3'-endo) sugar puckers, if that's what observed in the structure.

64. Line 193-195. Since G-1 is a ribonucleotide and G+1 is a deoxyribonucleotide, their typical sugar conformations should be North (C3'-endo) and South (C2'-endo), respectively. Are the authors sure that G-1 is in the South and G+1 in the North conformation in all structures? Fig. 3a shows that G-1 is indeed in the C2'-endo conformation; however the sugar conformation of G+1 in the green structure (methylated RNA) is unclear and probably wrong. Do the authors see same sugar puckers in the all-DNA structure as well as in the structure with a methylated substrate substitution (see my earlier comment)? If yes, this is a highly unusual observation and an interesting point to discuss. I am also not sure that the structures of DNAzyme and methylated DNAzyme determined at 3.05 and 3.8 Å resolution can tell about the sugar pucker. By the way, the leadzyme was crystallized with two different sugar puckers for the same residue.

65. Fig. 3. Large sticks for the highlighted nucleotides should be removed to simplify the view.

66. The authors did mention that a methyl moiety is not seen in the map in the Materials and Methods section; for those readers who do not read M&M section, this fact should be mentioned in the main text, possibly in the figure legend.

67. Line 198. In the proposed mechanism of the 8-17 DNAzyme ...

68. Line 203, Fig. 6. There is no need to show comparison with natural ribozymes in the main text figure. As expected, natural ribozymes differ from the in vitro selected DNAzyme.

69. Lines 206-209, SFig. 4 a,b. Parallels with the catalytic mechanism of the hammerhead and hairpin ribozymes are more interesting and can be presented in the main text figure. One thing is striking when the DNAzyme is compared to the hammerhead ribozyme: although both have a kink in the catalytic site, the interhelical angle is larger in the hammerhead ribozyme than in the DNAzyme. This observation relates to the question I've raised about smFRET data.

70. The authors can also compare their structure with the RNA-ligating DNAzyme structure; there are similarities which could be discussed.
71. SFig. 4. Why are labels shown in three different colors?
72. Line 213. Fig. 4e shows that lead-binding pocket does contain a charged residue. What is it?
73. Line 215. Does water displace O5' atom or donate a proton to this leaving group?
74. Line 241. Denaturing.

Reviewer #2 (Remarks to the Author):

The authors report the X-ray crystal structure of the RNA-cleaving 8-17 DNAzyme, a long-known member of the most-studied classes of DNAzyme. Many labs have tried for many years to obtain such a structure, so this manuscript will be viewed as a breakthrough, also because it is only the second structure (after ref. 9, Nature 2016) of any DNAzyme. The new 8-17 structure (which is actually three related structures) reveals several new and in some cases unexpected structural features, and the observed structure explains many chemical features of the catalysis.

After suitable revisions that do not appear to require any new experiments, the manuscript should be acceptable for Nature Communications.

(This reviewer is not a crystallographer and therefore leaves checking of the technical details of the crystallography to other reviewers who are expert on that aspect of the work.)

1. Page 4, line 83: "To unravel the catalytic mechanism of the DNAzyme, we determined three crystal structures...". However, the nature of these three structures (why three and not just one) is not revealed until page 6, line 115: "Among the three structures (DNAzyme without Pb²⁺, DNAzyme with Pb²⁺, and DNAzyme with O2'-Me-G substrate)...". The nature of the three structures should be mentioned on page 4 rather than waiting until page 6, especially because structural information is shown well before page 6 is reached. Whether the 2'-OMe-G structure was in the presence or absence of Pb²⁺ should also be made clear.
2. The relevant panels of Figure 1, and its caption, do not state that the structure shown was obtained in the presence of Pb²⁺, and Pb²⁺ is not depicted in any of the panels. Same for Figure 2.
3. Page 7, lines 139-143: this text does not account for the fact that methylation of guanosine O6 also disrupts the functional group at N1. Therefore, concluding that "the N1 site is more critical than the O6 site" may be unwise, also because the N1 methylation introduces a large group (methyl) in a different physical position than O6 methylation.

4. Page 10, line 202: "the overall fold and the detailed catalytic mechanism of 8-17 DNAzyme are completely different from the ribozymes (Fig 6)". However, the text also notes that each of 8-17 DNAzyme, hammerhead ribozyme, and hairpin ribozyme use a G residue as general base for deprotonation of the 2'-OH at the cleavage site. This aspect at least is not "completely different" among the DNAzyme and ribozymes (curiously, HDV is not mentioned at all here, although it is shown in the figure). I agree that the general acid aspect for protonation of 5'-leaving group is completely different.

5. Related to previous comment, and perhaps confusingly, the Conclusions (page 11 line 225) emphasizes "Our crystal structures have highlighted that similar to its counterpart ribozymes, this DNAzyme catalyzes the RNA cleavage via a general acid-base mechanism". So one part of this manuscript emphasizes "completely different" mechanisms, whereas another part "similar to ribozymes". This seems inconsistent.

6. Page 6, line 126: "2'-Me protection" should be "2'-OMe protection". Similarly, page 6, line 115 should be "2'-OMe-G substrate" rather than "O2'-Me-G substrate", if only to avoid the implication of a 2'-Me rather than 2'-OMe group. Note Figure 3 caption already says "DNAzyme(2'-OMe-G)".

7. In the Methods, the very brief description (page 13, line 258) that "the AsfvPolX protein is expressed and purified in the laboratory" is insufficient to allow others to reproduce the work. Was there an expression plasmid; if so, how was it prepared or from where was it obtained? What was the procedure for protein expression and purification?

8. Figure 4c, perhaps the water molecule can be labeled explicitly in the figure panel.

9. Figure 5 caption includes explanation of the D, S, and P labels. Such explanation should be provided for Figure 1b as well.

10. The manuscript would benefit from revision for grammar and spelling.

We sincerely thank both reviewers for reading our manuscript with great care, we also thank the reviewers for their very helpful comments, encouragements and criticisms as well. Based on these comments and suggestions, we have carefully revised our manuscript with the major changes highlighted in red in the revised manuscript. We believe that the quality of our manuscript has been significantly improved. The following are our point-to-point responses to the reviewers' comments.

Reviewer #1 (Remarks to the Author):

The study by Liu and coworkers is focused on determining the crystal structure and elucidating the catalytic mechanism of the RNA cleaving DNAzyme "8-17". The authors determined the 2.55 Å X-ray crystal structure of a sibling of the famous 10-23 DNAzyme and showed how a small DNA motif forms an amazing functional unit. This structure, to my knowledge, is the first structure of an RNA-cleaving DNAzyme. Most importantly, these DNAzymes are not some kind of oddity but are important tools used, for example, for fragmentation of natural mRNAs for subsequent analysis (see a recent example in Luciano et al, 2017, Mol. Cell). They also have potential for silencing harmful genes in mammals. The authors are very fortunate to obtain the structure in the conformation that appears to reflect a relevant pre-catalytic state. The structure, structure-guided mutagenesis and nucleotide substitution analysis provide a plausible mechanism for catalysis and explain the role of various parts of the DNA in the structure formation and catalysis. In addition, the structure visualized the bound cofactor, a lead cation. I find this study of great interest to general audience of Nature Communications and strongly suggest considering the manuscript for publication after a major revision.

Response: We sincerely thank the reviewer for the good comments and encouragements.

The manuscript has some deficiencies, which have to be addressed prior to publication.

The main critique is that the functional assays are not quantitative. Although the gels showing cleavage with different mutations and substitutions provide nice side-by-side comparisons of DNAzyme activity, I would expect the 21st century mechanistic study to rely on measured rate constants and not on qualitative descriptions such as “nearly abolished the enzymatic activity”, “almost completely abolished the activity”, “note as dramatic as”, “significantly reduce the activity”, etc. Based on these gels, I disagree with some of authors’ conclusions, and determination of rate constant values will resolve the controversy. Since the authors already have materials and the assay in hand, these experiments should take a couple of weeks.

Response: We sincerely thank the reviewer for the criticisms and very helpful suggestions. To better understand the function and the catalytic mechanism of 8-17 DNAzyme, we have redone most of the cleavage assays using FAM-labelled substrates. All the results have been quantified and included in the revised manuscript.

I also do not understand the structural basis for the lead cation selectivity and would feel more comfortable about the proposed mechanism of catalysis with less disruptive nucleotide substitutions (probably hard to do) and analysis of the published biochemical data based on structural observations. The authors also did not specify clearly that their structure is all-DNA, and not a DNAzyme bound to the RNA substrate, and that the structure represents one of the variants, albeit the most efficient, of the 8-17 DNAzyme. Overall, the manuscript would benefit from more clarity and comparison with a relevant

structure of the leadzyme and not distinct structures of some ribozymes.

Response: We sincerely thank the reviewer for the very helpful comments. We have included many details, such as crystallization, the DNAzyme and substrate compositions, in the Results section of the revised manuscript. As suggested by the reviewer, we also compared our structure with the leadzyme structure. We prefer to keep the comparison between our structure and other RNA-cleaving ribozymes, especially the hammerhead and hairpin ribozymes, which share some similarity in the catalytic mechanism with the DNAzyme.

The DNAzyme crystals are very fragile and they decay very fast during the data collection. To obtain the diffraction data for the three structures reported in the manuscript, we screened hundreds of crystals. We are really sorry that we could not obtain useful crystals and solve the structure containing less disruptive nucleotide substitutions, as suggested by the reviewer.

Specific critiques and suggestions to improve the manuscript are following.

1. Line 41-42. Please cite the original publications and not follow-up studies in this context.

For example, refs. 2-5 should be replaced by the original findings, refs. 13, 21, 22, etc.

Response: Done as suggested.

2. Line 46-48. In addition to the potential use for silencing genes in vivo, RNA cleaving

DNAzymes, including 10-23 enzyme, have other applications (see my comment earlier).

Response: We sincerely thank the reviewer for the very helpful comment. We have mentioned the mRNA fragmentation by DNAzyme and cited the wonderful reference in the revised manuscript.

3. The authors should carefully review all statements referring to prior studies to avoid factual errors. I have immediately noticed a few and suspect that there are more erroneous statements. For example, in lines 64-66, they cite ref. 21 published in 1997 and then say “later it turned out” and cite ref. 22, which was published in 1996.

Response: We are very sorry for the mistake and thank the reviewer for the helpful criticism. We have carefully re-written these statements referring to prior studies. Though Mg5 contains a motif similar to the 8-17 DNAzyme, its size (over 70 nt) is significantly longer than the canonical

8-17 DNAzyme. Therefore, we have deleted all the Mg5 related references in the revised manuscript.

4. Line 54. I would avoid citing ref. 22 about Mg5 enzyme and some other papers here and cite the original publication on the 8-17 DNAzyme.

Response: Done as suggested.

5. Line 63. Please provide the consensus sequences for the motif from the SELEX experiments in Fig. 1A. The crystallized variant belongs to one of the two consensus motifs described by Santoro and Joyce.

Response: The consensus sequence of 8-17 DNAzyme, which was adapted from the most commonly observed sequence variations and mutagenesis experiments, has been included in Figure 1a.

6. Line 66. The fact that the 8-17 DNA works better with transition metal cations is not surprising. There is a chemical explanation for this fact.

Response: Thanks for the correction. The sentence have been re-written in the revised manuscript.

7. Lines 83-84. Please spell out what was crystallized: a 36-mer DNA paired with a 23-mer substrate DNA (not RNA!) in the absence and presence of the cleavage cofactor, lead cation, as well as a DNA paired to a modified non-cleavable substrate that mimics the RNA substrate.

Response: We sincerely thank the reviewer for the very helpful comments. We have included one new paragraph at the beginning of Results section to summary the detailed components and crystallization of the three structures.

8. Line 85. Here, the overall structure could be first presented. I recommend showing the entire structure in Fig. 1. An image similar to SFig. 1a would be most useful. Please show nucleobases with sticks to simplify identification of nucleobases. Presentation in Fig. 1d does not let distinguish different purines. To simplify the view, sugars could be reduced to sticks. This image should be color-coded as in Fig. 1 b, with both P3 and P4 is respected colors. The authors may retain green color for the cartoon showing the backbone of the catalytic core but have nucleobases in cyan and green.

Response: We sincerely thank the reviewer for the great suggestions. The entire structure was depicted in the updated Fig. 1c; instead of cyan and green, the nucleobases of P3 and P4 were colored in blue and red, respectively, to achieve better resolution. The Fig. 1b was also updated accordingly in the revised manuscript.

9. Fig. 1c. The refined 2Fo-Fc electron density map does not illustrate the quality of the map. Please show the structure with an unbiased map (simulated annealing omit map) and move the figure to the Supplementary Figures section. Please retain the unbiased electron density map for the GG kink in the main text figure.

Response: The overall structure with an unbiased map was shown in the updated supplementary Fig. 1a. The unbiased electron density map for the GG kink was shown in Fig.2a in the revised manuscript.

10. Fig. 1d. With a new image of the structure, this panel might show zoomed-in view of the core, with less of P1 and P2. In addition, a “side” view (rotated 90 degrees along “y axis”) from the “P2 side” could help to understand the formation of the P3 “helix”. Please label P3 and P4.

Response: We sincerely thank the reviewer for the great suggestion. As suggested by the reviewer, the Fig. 1d has been replaced with a new image showing the zoomed-in view of the catalytic core in the revised

manuscript.

11. Fig. 1e. The authors highlight C7 and T11 here without explanations. The explanations are given in the second part of the write-up, which however does not refer to this figure.

Response: Fig. 1e has been replaced with a new image showing the detailed A5:G13 and G6:C12 pairing in the revised manuscript.

12. Fig. 1e. This view is difficult to understand because of the oversimplified presentation of the majority of nucleobases, shading of sugar rings, and insufficient transparency of overlapped nucleobases. Please add more labels for nucleotides.

Response: Fig. 1e has been replaced with a new image showing the detailed A5:G13 and G6:C12 pairing in the revised manuscript.

13. Fig.1 legend. “Globe” or “global” architecture?

Response: It is “global architecture”. Thanks for the correction.

14. Lines 92-93. I would expect a larger (closer to 180 degrees) angle between the arms according to smFRET measurements of the Pb²⁺-induced cleavage (Kim et al, 2007, NCB, 3: 763). Is crystal packing affect the angle? Can the angle get smaller without disruption of the structure? Please discuss this issue.

Response: We thank the reviewer for the very helpful comments and the useful literature. Using the smFRET method, Kim and the coworkers demonstrated that Pb²⁺ does not cause obvious conformational change during the cleavage reaction, this observation is consistent with our structural observation. In the literature, the authors also provided a schematic figure showing a larger (closer to 180 degrees) angle between the arms, but they did not provide any atomic structure model of the DNAzyme. Therefore, it is very difficult for us to compare our structure with the schematic model proposed by Kim et al.

At the beginning of the “the overall folding of 8-17 DNAzyme”

section of the revised manuscript, we discussed about the space groups and molecular packing of the three structures, which indicate that the presence of AsfvPolX has little effect on the folding of the core region of the DZ23/substrate complex and our structure should represent a real model of the complex.

15. Lines 98-100. In Fig. 1, G13 and A5 look like they are not in the same plane while G13 seems to be in the same plane with C12 and G6.

Response: G13 and A5 are in the same plane. In the revised manuscript, we have replaced the original Fig. 1e with a new figure to show the relative orientations of the four residues.

16. Line 97. C4:G8.

Response: Thanks for the correction.

17. Line 106. What does “non-canonical T1:G+1 wobble pair” mean? This is a “canonical” wobble base pair.

Response: Thanks for the correction. The “non-canonical T1:G+1 wobble pair” has been replaced with “canonical T1:G+1 wobble pair” in the revised manuscript.

18. Fig. 2a. This panel shows a surface view which was not discussed in the manuscript.

Response: Fig2a has been updated. The surface view was removed to better show the electron density of G-1 and G+1 residues.

19. Fig. 2b. Please label interacting nucleotides.

Response: Done as suggested.

20. Line 109. ... pseudoknot that resembles the shape of an inverted cone.

Response: Thanks for the suggestion.

21. Fig. 3 could be moved to the Supplementary Figures section. Please provide rmsd for the cores of the superposed structures.

Response: Fig. 3 was deleted in the revised manuscript. Instead, we included a new panel in the Supplementary Fig. 1c, showing the superposition of the DNAzyme-Pb²⁺ and DNAzyme(2'-OMe-G) structures.

22. Line 123. Please provide ref(s) for "early finding".

Response: The reference has been included in the revised manuscript.

23. Line 126-128. This section is misleading. It should be clearly explained that, first, the authors have crystallized the DNAzyme bound to a non-hydrolysable substrate strand made of DNA. Second, they wanted to introduce a ribonucleotide into the DNA strand to obtain a hydrolysable substrate but instead they incorporated a methyl-2'O substituted nucleotide to prevent possible cleavage. The structure was however obtained without Pb, therefore no cleavage was expected. I am wondering why the authors did not obtain the structure with a ribonucleotide in the absence of lead or the structure of the methylated substrate with lead. The structure of the methylated RNA substrate with lead would most closely represent a pre-catalytic state of the enzyme, but this structure was not obtained for unclear reasons. Right now, the catalytic mechanism model represents a combination of information from three structures and none of the structures individually corresponds to a pre-catalytic state structure.

Response: We sincerely thank the reviewer for the very helpful comments. We have included one new paragraph at the beginning of the Results section to describe the detailed components of the structures.

Besides the three structures reported in the manuscript, we tried many combinations of the DNAzyme, substrate (native DNA, DNA containing single rG, and DNA containing 2'-OMe-G), and cofactor (with or without

Pb²⁺). ***Though we got some crystals for the samples containing rG without Pb²⁺ or 2'-OMe-G with Pb²⁺, the crystals diffract too weak (typically lower than 7-8 Å) to give any useful information.***

24. Lines 128-130. There must be changes in the conformation of the GG kink, at least in the sugar-phosphate backbone. Please describe these changes, with emphasis on sugar conformations, and show a zoomed-in view of the differences.

Response: We sincerely thank the reviewer for the very helpful comments. We have carefully re-examined our structures, and as speculated by the reviewer, we found some conformational differences at the cleavage sites, which has been described under the "Conformational changes of the catalytic site residues" section of the revised manuscript.

25. Lines 130-134 and Fig. 4A. It is very hard to understand from Fig. 4a whether the alignment is indeed in-line or similar to in-line, as was reported earlier in the leadzyme ribozyme structure. Please make a better presentation of this important part, label the atoms involved in the alignment, show the near 180 degree angle (what is the angle value?), and show a zoomed-in view of the unbiased electron density map for the G-1 sugar and the adjacent atoms of the backbone to prove the alignment. Does the G-1 sugar adopt the C3' endo conformation?

Response: In the DNAzyme(2'-OMe-G) structure. the nucleophile (O2' atom of G-1), the electrophilic center (P atom of 3' phosphate of G-1), and the leaving group (O5' atom of G+1) adopt an almost in-line arrangement, the O2'-P-O5' angle is 160°. The three atoms have been labelled in the figure, which has been renumbered as Fig. 3a, in the revised manuscript.

The unbiased electron density map and the close-view of the cleavage site structure were shown in the Supplementary figure 2. Instead of C3'-endo, the sugar pucker of G-1 adopts C2'-endo conformation in the DNAzyme(2'-OMe-G) structure, which is similar to the G-1 sugar pucker observed in the DNAzyme-Pb²⁺ structure.

26. Line 133. Please split the sentence. Atoms of G13 are not involved in the in-line alignment.

Response: Done as suggested.

27. Line 135-142. The RNA field experienced many issues when relating ribozyme structures and catalytic mechanisms. Please use very careful wording discussing the catalytic mechanism of the 8-17 DNAzyme. Addition of a methyl moiety to the Watson-Crick edge of G13 can disrupt the structure and indirectly affect catalysis. This is what probably happens with the Dz36-6mG13 enzyme.

Response: We sincerely thank the reviewer for the very helpful comments. We have re-written the statement related to the catalytic mechanism in the revised manuscript. We are totally agreeing with the reviewer that introducing of a methyl moiety to the Watson-Crick edge of G13 could disrupt the structure and indirectly affect catalysis; this is consistent with our cleavage assay. To further support the functional role of the G13 residues, we included one new figure (Figs. 6e-6f) showing the similar arrangement of the catalytic G residue and cleavage site kink residues in the DNAzyme structure and the hammerhead structure, whose catalytic mechanism has been well characterized.

28. Lines 144-150, 158-169, Fig. 4c,e. Please show the Pb²⁺ binding site in more detail, with distances to heteroatoms of RNA. It is hard to imagine that a specifically bound cation nicely sitting in the cavity forms only a single coordination bond and nothing else participates in the cation binding. Can Pb²⁺ cation coordinate to N7 of G6? Other metal cations were found interacting with N7 of purines.

Response: We have carefully re-examined our structures; as depicted in the Supplementary Figs. 4a-4b, the distances between Pb²⁺ and the heteroatoms of the surrounding residues are within the range of 5-7 Å. In

contrast to the two-coordinated Pb^{2+} , these observations suggest that the catalytic core can accommodate Pb^{2+} with multi-coordination. However, as indicated by the occupancy (40%) of the Pb^{2+} , binding of the Pb^{2+} is very dynamic, which may lead to the disordering of some Pb^{2+} -coordinating water molecules that were not observed in the structure. *We sincerely thank the reviewer for the great comments and we have re-written this paragraph in the revised manuscript.*

Though the N7 atom of purines can coordinate with metal cations in some nucleic acid structures, however, as indicated by the long distance (4.0 Å) between them, the N7 atom of G6 does not coordinate with the Pb^{2+} ion in the DNAzyme structure.

29. Please show a lead cation in “real size” (~3.0 Å diameter) in Fig. 4 to help better understand space requirements.

Response: We thank the reviewer for the very helpful suggestion. A lead cation with diameter of 3.0 Å was shown in the updated Fig. 3e in the revised manuscript.

30. Line 145. Why is coordination of a Pb^{2+} cation to water unexpected? Metal-water coordination is often involved in catalytic reactions.

Response: Thanks for the correction.

31. Line 148-150. Please tone down this statement. The water molecule might be critical; however, this conclusion is based only on a sole H-bond distance.

Response: We sincerely thank the reviewer for the very helpful suggestion. The statement has been re-written in the revised manuscript.

32. Lines 151-156. Again, as in the earlier comment, these nucleotide changes may disrupt the structure and, without experimentally determined structures for mutant DNAzymes, conclusions must be carefully worded.

Response: We sincerely thank the reviewer for the very helpful suggestion. The statement has been re-written in the revised manuscript.

33. Lines 163-165. The reasons for higher catalytic activity in the presence of lead is not clear. Does a lead cation bind the enzyme better than other cations or is it better at the chemical step?

Response: We sincerely thank the reviewer for the very helpful comment. We believe that the higher catalytic activity of Pb^{2+} ion is a comprehensive result; in addition to its higher binding affinity with the DNAzyme, the low pKa value of Pb^{2+} ion may also contribute its higher catalytic ability. The statement has been re-written in the revised manuscript.

34. Does comparison of the DNAzyme activities with different cations correspond to the reverse order of pKa of the corresponding metal hydrates? What's a pH dependence of the cleavage rate with lead? Please discuss prior publications focused on the reaction chemistry in the context of the structure. Can all published biochemical data be explained by the structure-based catalytic mechanism?

Response: As reported by Brown and coworkers (Biochemistry, 2003, 42:7152-7161), the DNAzyme activity does not completely correspond to the reverse order of pKa of the cations. In the same literature, it was shown that the Pb^{2+} -dependent activity of DNAzyme increases linearly with pH and the slope of the plot of $\log k_{obs}$ versus pH is ~ 1 .

Unlike Pb^{2+} , previous data showed that the addition of Mg^{2+} , Zn^{2+} or some other cations will cause significant conformational change of the DNAzyme. We are very sorry that we could not solve any DNAzyme structure complexed with Mg^{2+} , Zn^{2+} or other cations; without these complex structures, we could not discuss and explain all the reported biochemical data, especially these data supported by Mg^{2+} or Zn^{2+} .

Upon carefully examination of previous literatures, we found that the effects of cations on the catalytic activity of DNAzyme have been well documented. Therefore, to keep our manuscript more concise and to avoid redundancy with previous literatures, we have deleted the in vitro cleavage assays using different cations in the revised manuscript.

35. Lines 163-165. I do not follow authors' idea and I do not see any structural reason for a lead cation to be a specific binder. Do the authors mean that a shallow cavity is too big for smaller than Pb^{2+} cations or too small for water-coordinated cations (such as fully hydrated Mg^{2+} cation) so that they cannot form a "productive" contact with an active water molecule and bind to the other "sides" of the cavity, away from the catalytic site? There are many precedents when cations such as Mg^{2+} or K^+ bind nucleic acids without coordinated water molecules.

Response: We sincerely thank the reviewer for the very helpful comments and we are totally agreeing with the reviewer that some cations can coordinate with different number of water molecules. Per the reviewer's suggestion, we have further analyzed our structure. As depicted in the Supplementary Figs. 4a-4b, the distances between Pb^{2+} and the heteroatoms of the surrounding core residues are within the range of 5-7 Å. Though only one Pb^{2+} -coordinating water molecule was observed in the structure, we believe that the DNAzyme core could accommodate Pb^{2+} ion coordinated with multiple water molecules.

As revealed by the leadzyme and the HDV ribozyme structures, cation recognition normally involves the phosphate backbone; however, due to the unique folding, the core region phosphate groups are not suitable for direct cation coordination in the DNAzyme structure. We believe that, instead of the space availability, lack of direct phosphate group coordination discriminates many cations from Pb^{2+} , which is flexible in coordination. Based on above analysis, we have re-written the related statement in the revised manuscript.

36. It is surprising that, having the crystals without bound cations in hand, the authors did not soak other cations, such as Mg^{2+} or its analogs with stronger anomalous scattering properties (Mn^{2+} , Ca^{2+}), to map the binding sites for these cations and put the issue to rest.

Response: We sincerely thank the reviewer for the very helpful

suggestions. In fact, we did the soaking experiment previously; however, very unfortunately, the DNAzyme crystals are very fragile and all the crystals cracked or lost diffraction power upon soaking. Therefore, we could not solve any DNAzyme structure complexed with Mg^{2+} , Mn^{2+} , or other cations.

37. Does Co hexamine, a mimetic of a fully hydrated Mg^{2+} cation, support the reaction?

Response: Yes, Co hexamine can support the cleavage reaction. However, as depicted in the figure below, the Co^{2+} -dependent activity of the DNAzyme is lower than the one supported by Pb^{2+} . To keep our manuscript more concise and to avoid redundancy with previous literatures (Biochemistry, 2003, 42:7152-7161), this result was not included in the revised manuscript.

38. The related DNAzyme 17E (Li, 2000, NAR, 28: 481-488) shows reduced activity in the presence of 150 mM F^- anions, consistent with the possibility that a fluoride can replace an active water molecule (as shown for some protein enzymes); however, the reaction is not abolished, arguing against a water molecule playing a critical role in the catalysis.

Please comment on this observation.

Response: We thank the reviewer for the very helpful comment. The reduced activity was observed in the presence of both Zn^{2+} and F^- . As discussed by Li and coworker in the literature, the binding affinity between Zn^{2+} and F^- is very strong. Instead of replacing and arguing against a catalytic water molecule for its critical role in the catalysis, we believe that F^- reduces the DNAzyme's activity through competing for the Zn^{2+} cofactor.

39. The proposed catalytic mechanism requires deprotonation of N1 of G13. This has not been discussed. What would shift the pKa of this group?

Response: As revealed by Brown and coworkers (*Biochemistry*, 2003, 42:7152-7161), the DNAzyme has very strong pH correlation; compared to the lower pH condition, it is more active under the higher pH condition. Based on these observations, we believe that, most likely, the N1 is deprotonated by a hydroxide anion existing in the environment; similar mechanism has been previously proposed for the hammerhead ribozyme, which also utilizes one G residue as the general acid (Matrick and Scott, *Cell*, 2006, 126(2), 209-320).

40. Is there anything to stabilize the transition state?

Response: We believe that there might have some interactions that can stabilize the transition state. However, without a structure representing the transition state, it is difficult for us to predict the detailed interaction.

41. Does Ba²⁺ cation, the most close mimetic of Pb²⁺ cation, support the reaction?

Response: Ba²⁺ can support the cleavage reaction. However, as depicted in the figure below, the Ba²⁺-dependent activity of the DNAzyme is much lower than the one supported by Pb²⁺. To keep our manuscript more concise and to avoid redundancy with previous literatures (*Biochemistry*, 2003, 42:7152-7161), this result was not included in the revised manuscript.

42. What's the occupancy of Pb²⁺ in the structure?

Response: Might due to the dynamic binding between the DNAzyme and the Pb²⁺, the Pb²⁺ is 40% in the structure.

43. Please show an omit and anomalous maps for Pb²⁺ cation. The authors must prove the identity of lead because the lead-containing structure was determined at higher resolution than the other structures and it has 20 water molecules emerged because of improvement of resolution.

Response: *The omit map of Pb²⁺ was depicted in Fig. 3c in the revised manuscript. We are very sorry that the DNAzyme-Pb²⁺ structure was solved by molecular replacement method; the diffract data was not collected at the peak wavelength of Pb²⁺ and it could not provide enough anomalous signal to generate the anomalous maps for Pb²⁺.*

Recently, we tried to re-collect the diffraction data at the peak wavelength of Pb²⁺. Though we screened many crystals, we failed to get any useful data, due to the fragility and quick decay of the crystals.

44. Please use the same color for lead cation throughout the figures. In Fig. 4, lead shown in black, grey and green colors, with labels in black and green.

Response: *Thanks for the suggestion. In the revised manuscript, the Pb²⁺ ion was colored in black in all the figures and labelled in black in the updated Fig. 3, which is corresponding to the original Fig. 4.*

45. Crystallographic table does not list B-factors for DNA, metal and water molecules.

Response: *All the B-factors for DNA, metal and water molecules have been included in the crystallographic table.*

46. Fig. 4e. A standard blue-red presentation of the surface potential could be a better option for this panel.

Response: *Thanks for the suggestion. We tried to show the surface potential using the standard blue-red presentation; however, as depicted in the figure below, it's very difficult to tell the oxygen atoms of the nucleobases and sugar puckers from those of the phosphate groups, which are negative in charge. Therefore, we prefer to keep the original color in the figure, to better show the relative orientations of the Pb²⁺ ion and the oxygen atoms of the phosphate groups.*

47. Line 483. Why is the water molecule shown in cyan and not in standard red color for an oxygen atom, with density in green or blue? What's the B-factor of this water molecule?

Response: The figure has been updated with the water molecule and the electron density colored in red and green, respectively. The B-factor of the water molecule is 51.

48. Fig. 4d. G+1 is a deoxyribonucleotide and therefore it should not have a 2'-OH group unless the authors say in the figure legend that they are showing all-RNA substrate.

Response: Thanks for the very helpful comment. We have re-written the figure legend to indicate that the mechanism is proposed for the RNA substrate cleavage by the DNAzyme.

49. Fig. 4d. Why are labels shown in two colors?

Response: The labels were all colored in black in the figure, which has been renumbered to Fig. 3d in the revised manuscript.

50. Please compare the structure and the putative catalytic mechanism of the DNAzyme with the structure and catalysis of the leadzyme.

Response: The overall folding and the catalytic site architecture of the DNAzyme and leadzyme have been compared in the revised manuscript. Though several NMR (Hoogstraten, J. Mol. Biol, 1998, 284:337-350) and the X-ray (Wedekind, Nat. Struct. Biol., 1999, 6(3):261-268) structures of leadzyme have been reported, they all captured the leadzyme in the inactive ground state; to date, no active form leadzyme structure has been reported and the detailed catalytic mechanism of the leadzyme remains elusive. Without an active form leadzyme structure, it is difficult for us to compare the detailed catalytic mechanism of the leadzyme and the DNAzyme.

51. Line 173. ...activity measurements...

Response: Thanks for the correction.

52. Fig. 5a. This figure is very crowded and unclear. See my earlier comments for Fig. 1e

to improve the view.

Response: Thanks for the great suggestion. This figure has been updated in the revised manuscript.

53. Fig. 5a. Is T11 cyan or dark blue? What' the magenta nucleotide?

Response: Fig. 5a has been updated and renumbered to Fig. 4a in the revised manuscript. In the updated figure, both T11 and C7 were colored in yellow. The magenta nucleotide is A14, which was colored in green and labeled in the updated figure.

54. Fig. 5b,c. Motivation for showing the electron density map is not clear. These are not

the most important regions of the DNAzyme and the refined 2Fo-Fc map is not the best

way to illustrate the quality of the structure.

Response: Thanks for the helpful comments. The electron density map has been removed from the figures, which have been updated and renumbered to Fig. 4b and 4c in the revised manuscript.

55. Lines 177-180 and 182-184. Motivation for testing an insertion of nucleotides at position 7 (4 mutants in total) and conclusions are not obvious. What do these mutations address? C7 provides spacing between adjacent base pairs and the structure shows that there is enough space for looping out a residue without impact on catalysis unless the inserted base is capable of forming alternative base pairs and disrupting the fold. That's what the authors see: insertion of purines that have better potential for base pairing is more disruptive than insertion of pyrimidines.

Response: We sincerely thank the reviewer for the very helpful comments. To keep the manuscript more concise, the DNAzyme mutants with nucleotide insertion at position 7 and the related in vitro cleavage assay results have been deleted from the revised manuscript.

56. Lines 180-182. This is an incorrect conclusion. While deletion of T11 does significantly reduce cleavage, insertions do not affect cleavage efficiency strongly, leading to the same conclusion as before: an insertion can be tolerated with only small impact on activity.

Response: Similar to the mutants with nucleotide insertion at position 7, the mutants with nucleotide insertion at position 11 and the related in vitro cleavage assay results were also deleted from the revised manuscript, to keep the manuscript more concise.

57. Line 185. Presented figures do not help to evaluate the potential role of A15 and A14.

Response: To better show the conformation of A14 and A15, a new figure (Fig. 5a) was included in the revised manuscript.

58. Lines 185-187. If I understand correctly authors' idea, deletion of A15 converts the 5'-WCGAA consensus sequence into the 5'-WCGR sequence, both observed in the original publication (Santoro et al). This result means that both consensus sequences from

SELEX correspond to the same DNAzyme structure.

Response: Yes, the mutant with A15 deletion was designed to confirm that DNAzymes with 5'-WCGR or 5'-WCGAA sequence share similar structures and catalytic activities.

59. Line 189. There is no Fig. 3c in the paper.

Response: We are so sorry for the mistake.

60. Lines 188-190. This conclusion is not entirely correct. According to the original observation (Santoro et al) and SFig. 3A, the A14G substitution in the context of the delA15 shows some activity. Same is true for A14T. This means that the A14:G-1 pair is important but not critical for catalysis.

Response: Thanks for the very helpful comments. We are totally agreeing with the reviewer that A14:G-1 pair is important but not critical for catalysis, the corresponding conclusion has been re-written in the revised manuscript.

61. Lines 190-193. This sentence is also not entirely accurate. While the majority of Watson-Crick pairs replacing the A14-G-1 base pair show diminished activity, the T-rA combination is rather active (SFig. 3b) and several non-canonical base pairs, A-rA (SFig. 3a), A-rC (SFig. 3d), T-rC (SFig. 3d) and T-rU (SFig. 3c) also show good activity. The authors cannot make strong conclusions without measured rate constants. It is recommended to provide a supplementary figure with a structure-based schematic of these combinations and discuss similar data from prior publications.

Response: We sincerely thank the reviewer for the very helpful comments and suggestions. We have redone the cleavage assays using FAM-labeled substrates and the results have been quantified in the

revised manuscript. Our results indicate that 8-17 DNAzyme prefers the purine substrates with a non-Watson-Crick paired combination (especially the A-rG pair) at the catalytic site. May due to the formation of stable Watson-Crick C:G pair, the SrG substrate cleavage activity of Dz35-C is much lower than other mutants. In contrast to the C-rG combination, Dz35-T is quite activity toward the SrA substrate. Compare to the C:rG pairing, the T:rA pairing is more dynamic. We believe that the dynamic T:rA pairing offers the Dz35-T mutant with good SrA cleavage activity, however, we don't have a structure to verify this hypothesis. Therefore, we could not discuss these base combinations in more details in the manuscript; along the same line, we are not confident to draw a schematic figure to summarize the base combinations in the revised manuscript.

62. Lines 193-195. The kink is usually stabilized through extensive stacking interactions and that's what the structure shows. The identity of base pairs, Watson-Crick or non-canonical, for making a bent in the backbone should not matter. What is likely important is that non-canonical base pairs are more dynamic than canonical base pairs and therefore offer flexibility required for the catalytic reaction. Published articles have discussed this point and the authors may discuss it from the structural perspective as well.

Response: Thanks for the great comments. In the revised manuscript, we carefully compared our structure with some RNA-cleaving ribozymes, especially the hammerhead ribozyme that shares many similarities with 8-17 DNAzyme. Based on the structural comparison, we discussed the kinking and non-canonical pairing at the end of the Results section.

63. Line 195. Please provide canonical designations of the South (C2'-endo) and North (C3'-endo) sugar puckers, if that's what observed in the structure.

Response: Thanks for the suggestion. This sentence has been deleted in the revised manuscript. The detailed conformations of G-1 and G+1 sugar puckers were described in other section; the canonical designations of the South and North sugar puckers were given in the revised manuscript.

64. Line 193-195. Since G-1 is a ribonucleotide and G+1 is a deoxyribonucleotide, their typical sugar conformations should be North (C3'-endo) and South (C2'-endo), respectively. Are the authors sure that G-1 is in the South and G+1 in the North conformation in all structures? Fig. 3a shows that G-1 is indeed in the C2'-endo conformation; however the sugar conformation of G+1 in the green structure (methylated RNA) is unclear and probably wrong. Do the authors see same sugar puckers in the all-DNA structure as well as in the structure with a methylated substrate substitution (see my earlier comment)? If yes, this is a highly unusual observation and an interesting point to discuss. I am also not sure that the structures of DNAzyme and methylated DNAzyme determined at 3.05 and 3.8 Å resolution can tell about the sugar pucker. By the way, the leadzyme was crystallized with two different sugar puckers for the same residue.

Response: We sincerely thank the reviewer for the very helpful comments. We are sure about the C2'-endo and C3'-endo conformations of G-1 and G+1 in the all-DNA structure. However, as pointed out by the reviewer, the resolution of the methylated DNAzyme structure is relatively low (3.8 Å), we are not absolutely sure about the conformations of the sugar puckers of G-1 and G+1 in this structure; the conformations of G-1 and G+1 sugar puckers were judged by the refinement program. Due to this uncertainty, we could not further discuss the sugar pucker conformations and compare them with other structures, including the leadzyme.

65. Fig. 3. Large sticks for the highlighted nucleotides should be removed to simplify the view.

Response: The original Fig. 3. has been deleted in the revised manuscript.

66. The authors did mention that a methyl moiety is not seen in the map in the Materials

and Methods section; for those readers who do not read M&M section, this fact should be mentioned in the main text, possibly in the figure legend.

Response: Thanks for the suggestion. Disordering of the methyl moiety has been mentioned in the main text.

67. Line 198. In the proposed mechanism of the 8-17 DNAzyme ...

Response: Thanks for the suggestion.

68. Line 203, Fig. 6. There is no need to show comparison with natural ribozymes in the main text figure. As expected, natural ribozymes differ from the in vitro selected DNAzyme.

Response: This figure has been moved to the Supplementary Figures section.

69. Lines 206-209, SFig. 4 a,b. Parallels with the catalytic mechanism of the hammerhead and hairpin ribozymes are more interesting and can be presented in the main text figure. One thing is striking when the DNAzyme is compared to the hammerhead ribozyme: although both have a kink in the catalytic site, the interhelical angle is larger in the hammerhead ribozyme than in the DNAzyme. This observation relates to the question I've raised about smFRET data.

Response: Thanks for the very helpful comments. SFig. 4 has been moved to the main text in the revised manuscript. Per the reviewer's suggestion, we have further compared the DNAzyme and hammerhead ribozyme structures. As depicted in the supplementary figure 7f, though the overall folding of the two structures are very different, the conformations of the substrate strands, especially the kinks and the flanking regions, are very similar in the two structures. Instead of the

linear like conformation proposed from the smFRET data, these structural similarities further support the DNAzyme conformations observed in our crystal structures.

70. The authors can also compare their structure with the RNA-ligating DNAzyme structure; there are similarities which could be discussed.

Response: Thanks for the suggestion. We tried to compare our structures with the RNA-ligating DNAzyme structure (PDB_ID: 5CKK), however, we could not find obvious similarities between them, including the overall folding and the catalytic site architecture. Therefore, we did not compare these structures in the revised manuscript.

71. SFig. 4. Why are labels shown in three different colors?

Response: All the labels were colored in black in the updated figure, which has been moved to the main text and renumbered to Fig. 6.

72. Line 213. Fig. 4e shows that lead-binding pocket does contain a charged residue.

What is it?

Response: The charged residue is A5. However, as depicted in the Supplementary figure 4a, the phosphate group of A5 does not coordinate with the Pb^{2+} ion, indicating by the shortest distance (5.1 Å) between the Pb^{2+} ion and the oxygen atom of the phosphate group.

73. Line 215. Does water displace O5' atom or donate a proton to this leaving group?

Response: The water molecule donates a proton to the leaving group.

74. Line 241. Denaturing.

Response: Thanks for the correction.

Reviewer #2 (Remarks to the Author):

The authors report the X-ray crystal structure of the RNA-cleaving 8-17 DNAzyme, a long-known member of the most-studied classes of DNAzyme. Many labs have tried for many years to obtain such a structure, so this manuscript will be viewed as a breakthrough, also because it is only the second structure (after ref. 9, Nature 2016) of any DNAzyme. The new 8-17 structure (which is actually three related structures) reveals several new and in some cases unexpected structural features, and the observed structure explains many chemical features of the catalysis.

Response: We sincerely thank the reviewers for all the good comments and encouragements.

After suitable revisions that do not appear to require any new experiments, the manuscript should be acceptable for Nature Communications.

(This reviewer is not a crystallographer and therefore leaves checking of the technical details of the crystallography to other reviewers who are expert on that aspect of the work.)

1. Page 4, line 83: "To unravel the catalytic mechanism of the DNAzyme, we determined three crystal structures...". However, the nature of these three structures (why three and not just one) is not revealed until page 6, line 115: "Among the three structures (DNAzyme without Pb^{2+} , DNAzyme with Pb^{2+} , and DNAzyme with O2'-Me-G substrate)...". The nature of the three structures should be mentioned on page 4 rather than waiting until page 6,

especially because structural information is shown well before page 6 is reached.

Whether the 2'-OMe-G structure was in the presence or absence of Pb^{2+} should also be made clear.

Response: We sincerely thank the reviewer for the very helpful comments. We have included one new section "Crystallization and structural determination of 8-17 DNAzyme" in the revised manuscript, to summarize the details of crystallization and compositions of the DNAzyme and substrate in the structures.

2. The relevant panels of Figure 1, and its caption, do not state that the structure shown was obtained in the presence of Pb^{2+} , and Pb^{2+} is not depicted in any of the panels.

Same for Figure 2.

Response: We sincerely thank the reviewer for the very helpful comments. We have re-written the figure legend in the revised manuscript, to point out that the figure is based on the DNAzyme- Pb^{2+} structure. To keep the figure clean, the Pb^{2+} was not shown in Figs. 1 and 2, but it was shown in the Fig. 3 and the Supplementary figure 4.

3. Page 7, lines 139-143: this text does not account for the fact that methylation of guanosine O6 also disrupts the functional group at N1. Therefore, concluding that "the N1 site is more critical than the O6 site" may be unwise, also because the N1 methylation introduces a large group (methyl) in a different physical position than O6 methylation.

Response: Thanks for the very helpful comments. The related conclusion has been re-written in the revised manuscript.

4. Page 10, line 202: "the overall fold and the detailed catalytic mechanism of 8-17 DNAzyme are completely different from the ribozymes (Fig 6)". However, the text also notes that each of 8-17 DNAzyme, hammerhead ribozyme, and hairpin ribozyme use a G residue as general base for deprotonation of the 2'-OH at the cleavage site. This aspect at

least is not "completely different" among the DNAzyme and ribozymes (curiously, HDV is not mentioned at all here, although it is shown in the figure). I agree that the general acid aspect for protonation of 5'-leaving group is completely different.

Response: We sincerely thank the reviewer for the correction. We have carefully re-examined and compared our structure with the RNA-cleaving ribozymes. The related paragraphs have been re-written in the manuscript, to better summarize the similarities and differences between DNAzyme and the RNA-cleaving ribozymes.

5. Related to previous comment, and perhaps confusingly, the Conclusions (page 11 line 225) emphasizes "Our crystal structures have highlighted that similar to its counterpart ribozymes, this DNAzyme catalyzes the RNA cleavage via a general acid-base mechanism". So one part of this manuscript emphasizes "completely different" mechanisms, whereas another part "similar to ribozymes". This seems inconsistent.

Response: We sincerely thank the reviewer for the great comments. The related paragraphs have been re-written in the manuscript, to better summarize the similarities and differences between DNAzyme and the RNA-cleaving ribozymes.

6. Page 6, line 126: "2'-Me protection" should be "2'-OMe protection". Similarly, page 6, line 115 should be "2'-OMe-G substrate" rather than "O2'-Me-G substrate", if only to avoid the implication of a 2'-Me rather than 2'-OMe group. Note Figure 3 caption already says "DNAzyme(2'-OMe-G)".

Response: Thanks for the correction.

7. In the Methods, the very brief description (page 13, line 258) that "the AsfvPolX protein is expressed and purified in the laboratory" is insufficient to allow others to reproduce the

work. Was there an expression plasmid; if so, how was it prepared or from where was it obtained? What was the procedure for protein expression and purification?

Response: We sincerely thank the reviewer for the great comments. The detailed procedures for DNA construction, protein expression and purification of AsfvPoIX has been reported in our recent study (Plos Biol, 2017, 15(2): e1002599), which has been cited in our revised manuscript

8. Figure 4c, perhaps the water molecule can be labeled explicitly in the figure panel.

Response: Done as suggested.

9. Figure 5 caption includes explanation of the D, S, and P labels. Such explanation should be provided for Figure 1b as well.

Response: Thanks for the suggestion. All the DNazymes, substrates, and products have been indicated on the in vitro cleavage assay gel and explained in all the figure legends.

10. The manuscript would benefit from revision for grammar and spelling.

Response: We sincerely thank the reviewer for the great suggestion. We have carefully revised our manuscript, which was also polished by one language services company.

Reviewers' Comments:

Reviewer #1 (Remarks to the Author):

In the revised version of the manuscript, Gan and coworkers addressed all my comments and accommodated the majority of my suggestions. I find the revised version greatly improved and almost ready for publication. I support authors' decision to remove less relevant biochemical experiments & data and appreciate their efforts on quantifying reaction rates. In my opinion, the lead specificity remains puzzling and this work provides a good basis to address this interesting issue in future studies. There is one point, which I would like to clarify:

- (1) Point 43. I think the authors misunderstood my comment. I asked to show an anomalous map, not to solve a structure with the anomalous signal of lead. At 1.0 Å wave length, lead has strong anomalous signal ($\sim 5 e$), which is ~ 10 -fold stronger than the anomalous signal of other atoms in the structure. Even if the anomalous signal is not apparent during data processing, the anomalous signal of lead at 40% occupancy will be huge at the anomalous map. The authors should simply reprocess their existing data to include anomalous signal and build an anomalous map in COOT using the MR phases. Since Phenix retains DANO and SIGDANO columns, this should be very easy to do after refinement in Phenix. This could be a nice support for the identity of the cation. I understand that the cation identity was deduced by comparing the + and - Pb data; however given the large difference in resolution, it would be nice to have an additional confirmation.
- (2) Please double-check figure numbers. For example, in line 281, Fig. 4d must be Fig. 3d.
- (3) Line 283. I cannot understand how 2'-OH of G-1 displaces O5' of G+1. Perhaps this sentence should be re-worded.

Reviewer #2 (Remarks to the Author):

The authors have revised their manuscript to account for the comments of both reviewers (I was reviewer 2). They made many changes to the manuscript, and these changes appear to address comprehensively all of the prior review comments.

On line 276, "the pKa value of Pb²⁺" should refer instead to the pKa value of water coordinated to Pb²⁺, as the metal ion Pb²⁺ itself of course has no pKa value.

We sincerely thank both reviewers for encouragements and reading our manuscript with great care. We would also like to thank the reviewers for all their previous and current comments and suggestions, which have significantly improved the quality of our manuscript. The following are our point-to-point responses to the reviewers' comments.

REVIEWERS' COMMENTS:

Reviewer #1 (Remarks to the Author):

In the revised version of the manuscript, Gan and coworkers addressed all my comments and accommodated the majority of my suggestions. I find the revised version greatly improved and almost ready for publication. I support authors' decision to remove less relevant biochemical experiments & data and appreciate their efforts on quantifying reaction rates.

Response: We sincerely thank the reviewer for all the good comments and encouragements.

In my opinion, the lead specificity remains puzzling and this work provides a good basis to address this interesting issue in future studies. There is one point, which I would like to clarify:

(1) Point 43. I think the authors misunderstood my comment. I asked to show an anomalous map, not to solve a structure with the anomalous signal of lead. At 1.0 Å wave length, lead has strong anomalous signal (~5 e), which is ~10-fold stronger than the anomalous signal of other atoms in the structure. Even if the anomalous signal is not apparent during data processing, the anomalous signal of lead at 40% occupancy will be huge at the anomalous map. The authors should simply reprocess their existing data to include anomalous signal and build an anomalous map in COOT using the MR phases. Since Phenix retains DANO and SIGDANO columns, this should be very easy to do after refinement in Phenix. This could be a nice support for the identity of the cation. I understand that the cation identity was deduced by comparing the + and - Pb data; however, given the large difference in resolution, it would be nice to have an additional confirmation.

Response: We are so sorry for the misunderstanding and sincerely thank the reviewer for the very patient explanation. As suggested by the reviewer, we have reprocessed the data. As depicted in the table below, the Pb²⁺ ion does generate some weak anomalous signal at low resolution (about 5.5 Å); however, may due to the low occupancy of Pb²⁺

and the quick decay of the crystal, no clear peak was observed for the Pb^{2+} on the anomalous map.

SUBSET OF INTENSITY DATA WITH SIGNAL/NOISE ≥ -3.0 AS FUNCTION OF RESOLUTION

RESOLUTION LIMIT	NUMBER OF REFLECTIONS			COMPLETENESS OF DATA	R-FACTOR observed	R-FACTOR COMPARED expected	I/SIGMA	R-meas	CC(1/2)	Anomal Corr	SigAno	Nano
	OBSERVED	UNIQUE	POSSIBLE									
14.67	3260	463	465	99.6%	6.8%	6.6%	3260	27.04	7.4%	99.5*	12	179
10.47	5290	813	813	100.0%	6.2%	6.6%	5290	26.36	6.8%	99.7*	2	352
8.58	6903	1018	1018	100.0%	7.0%	6.8%	6903	25.10	7.6%	99.4*	5	458
7.44	8918	1252	1252	100.0%	7.7%	7.6%	8918	21.56	8.3%	99.5*	-3	571
6.66	10057	1371	1372	99.9%	8.6%	8.6%	10057	18.43	9.2%	99.5*	-1	631
6.09	10843	1546	1546	100.0%	9.3%	9.4%	10843	16.64	10.0%	99.3*	-7	717
5.64	10669	1660	1663	99.8%	10.0%	10.4%	10669	14.29	10.9%	99.2*	-8	774
5.27	12410	1795	1795	100.0%	11.1%	12.0%	12410	13.18	12.0%	99.2*	-6	841
4.97	13157	1887	1895	99.6%	11.8%	12.4%	13150	12.82	12.7%	99.1*	0	887
total	81507	11805	11819	99.9%	7.8%	7.9%	81500	17.71	8.4%	99.7*	-2	5410

Though the omit map, mutagenesis and in vitro cleavage assay all suggested that Pb^{2+} coordinates with G6 of 8-17 DNAzyme, we agree with the reviewer that the lead specificity needs to be further investigated. We have reflected this point in the manuscript. As commented by the reviewer previously, there are many questions remain to be answered, such as the binding of Mg^{2+} , Zn^{2+} , and Ca^{2+} with the DNAzyme. We will keep working on this project and we hope that more 8-17 DNAzyme structures with higher resolution and fully occupied cations could be solved in the near future, these structures will provide more information for the cation specificity of the enzyme.

(2) Please double-check figure numbers. For example, in line 281, Fig. 4d must be Fig. 3d.

Response: Thanks for the correction. As suggested by the reviewer, we have carefully examined all the figure numbers.

(3) Line 283. I cannot understand how 2'-OH of G-1 displaces O5' of G+1. Perhaps this sentence should be re-worded.

Response: We sincerely thank the reviewer for the helpful comment. The sentence has been rewritten in the revised manuscript.

Reviewer #2 (Remarks to the Author):

The authors have revised their manuscript to account for the comments of both reviewers (I was reviewer 2). They made many changes to the manuscript, and these changes appear to address comprehensively all of the prior review comments.

Response: We sincerely thank the reviewer for all the good comments and encouragements.

On line 276, "the pKa value of Pb²⁺" should refer instead to the pKa value of water coordinated to Pb²⁺, as the metal ion Pb²⁺ itself of course has no pKa value.

Response: Thanks for the correction.